# Hepatic transcriptome analysis identifies genes, polymorphisms and pathways involved in the fatty acids metabolism in sheep

**Asep Gunawan**[1], **Kasita Listyarini**[1], **Ratna Sholatia Harahap**[1], **Jakaria**[1], **Katrin Roosita**[2], **Cece Sumantri**[1], **Ismeth Inounu**[3], **Syeda Hasina Akter**[4,5], **Md. Aminul Islam**[4,6], **Muhammad Jasim Uddin**[4,5,7] *

1 Department of Animal Production and Technology, Faculty of Animal Science, IPB University, Bogor, Indonesia, 2 Department of Nutrition Science, Faculty of Human Ecology, IPB University, Bogor, Indonesia, 3 Center Research for Development of Animal Husbandry, Bogor, Indonesia, 4 Faculty of Veterinary Science, Bangladesh Agricultural University, Mymensingh, Bangladesh, 5 School of Veterinary Science, The University of Queensland, Gatton, Australia, 6 Food and Feed Immunology Group, Graduate School of Agricultural Science, Tohoku University, Sendai, Japan, 7 School of Veterinary Medicine, Murdoch University, Perth, Western Australia, Australia

* jasim.uddin@murdoch.edu.au

**Data Availability Statement:** All data are available in this manuscript. We used our originally generated transcriptome data set (RNA

## Abstract

Fatty acids (FA) in ruminants, especially unsaturated FA (USFA) have important impact in meat quality, nutritional value, and flavour quality of meat, and on consumer's health. Identification of the genetic factors controlling the FA composition and metabolism is pivotal to select sheep that produce higher USFA and lower saturated (SFA) for the benefit of sheep industry and consumers. Therefore, this study was aimed to investigate the transcriptome profiling in the liver tissues collected from sheep with divergent USFA content in longissimus muscle using RNA deep-sequencing. From sheep (n = 100) population, liver tissues with higher (n = 3) and lower (n = 3) USFA content were analysed using Illumina HiSeq 2500. The total number of reads produced for each liver sample were ranged from 21.28 to 28.51 million with a median of 23.90 million. Approximately, 198 genes were differentially regulated with significance level of p-adjusted value <0.05. Among them, 100 genes were up-regulated, and 98 were down-regulated (p<0.01, FC>1.5) in the higher USFA group. A large proportion of key genes involved in FA biosynthesis, adipogenesis, fat deposition, and lipid metabolism were identified, such as APOA5, SLC25A30, GFPT1, LEPR, TGFBR2, FABP7, GSTCD, and CYP17A. Pathway analysis revealed that glycosaminoglycan biosynthesis- keratan sulfate, adipokine signaling, galactose metabolism, endocrine and other factors-regulating calcium metabolism, mineral metabolism, and PPAR signaling pathway were playing important regulatory roles in FA metabolism. Importantly, polymorphism and association analyses showed that mutation in APOA5, CFHR5, TGFBR2 and LEPR genes could be potential markers for the FA composition in sheep. These polymorphisms and transcriptome networks controlling the FA variation could be used as genetic markers for FA composition-related traits improvement. However, functional validation is required to confirm the effect of these SNPs in other sheep population in order to incorporate them in the sheep breeding program.

sequencing). The data has been submitted to NCBI (Accession: PRJNA764003, ID: 764003). We validated this data in our laboratory and we performed the genotyng experiment in our study population of sheep. This is not a meta analysis.

**Funding:** This work was supported by a project World Class Research (WCR) Number: 077/SP2H/LT/DRPM/2021 from the Ministry of Education of the Republic of Indonesia.

**Competing interests:** The authors have declared that no competing interests exist.

## Introduction

Meat quality is an economically important trait because of consumer's choice which includes both visual and sensory traits, health benefits, and humane production system. Recently, fatty acids (FA) composition is being considering as a new feature for lamb quality [1]. Ruminants' meat is generally containing higher levels of saturated fatty acids (SFA), which are widely correlated with health problem such as heart disease, stroke, and obesity [2], so consumers are favouring leaner meats containing less SFA and higher polyunsaturated fatty acids (PUSFA) [3, 4]. PUSFA, mainly omega-3 are considered beneficial for human health that reduce the serum low density lipoprotein (LDL)-cholesterol, total cholesterol concentration, and modulate immune functions [5]. Additionally, desirable sensorial characisctic of meat is associated with PUSFA and MUSFA (monounsaturated fatty acids) [6]. Note, sheep meat is rich in omega-3 long-chain ($\geq$20) FA ($\omega$3 LC-PUSFA), eicosapentaenoic (EPA, 20:5$\omega$3), and docosahexaenoic (DHA, 22:6$\omega$3) which are beneficial for human health and immunity [7]. Meat production with a higher PUSFA and lower SFA content is, therefore, important to improve human health without requiring substatial changes in customers' habit of meat consumption.

Molecular breeding is recommended as one of the most realistic approaches for increasing PUSFA- and reducing SFA-content. However, identification of the candidate genes and genomic networks is the first step to achieve the goal. Notably, FA compositions are the well-defined compounds describing the phenotypic traits which are possible to improve through genetic selection. FA compositions show moderate to high heritability ranging from 0.15 to 0.63 [8, 9]. Identification of genetic factors controlling FA composition could be implemented in breeding programmes to select animals that produce higher PUSFA and lower SFA in meat. Therefore, it is crucial to understand the genomics of FA metabolism to select sheep with higher PUSFA and lower SFA content. FA metabolism is a complex process, which involves lipolysis of dietary fat, biohydrogenation in the rumen, and *de novo* synthesis of FA by rumen bacteria. Furthermore, absorption and transport of FA by the host animal, *de novo* synthesis, elongation and desaturation in the animal's tissues, hydrolysis of triglycerides, esterification, and the oxidation of FA or its metabolization into other components together make it a complex process to decipher [10].

High-throughput sequencing technologies (RNA-Seq) are now widely using for transcriptome analysis because of an unprecedented accuracy and data insight [11]. The reliable and comprehensive data from RNA-Seq can not only describe the genes' structure, but also provide a better understanding of the biological function of genes [12]. This technology is allowing the animal breeding industry to significantly increase the rate of genetic progress [13]. Several recent studies have used RNA deep sequencing to identify differentially expressed genes related to FA metabolism in muscle and liver in domestic animals such as in pigs [14, 15], and cattle [16]. But our understanding of genomic signature behind the FA metabolism in sheep at the molecular level is limited. Although several candidate genes, such as ACACA [17], FASN and SCD [18] are reported to be associated with FA and fat content in various sheep breeds, the whole genomics underlying the FA metabolism in sheep is remained to be deciphered. In accordance with other studies of FA composition, there is an inevitable need for using RNA deep sequencing for transcriptome profiling related to higher PUSFA and lower SFA in sheep. Therefore, the aim of this study was to elucidate the genes and pathways involved in FA metabolism in the liver tissue using RNA deep sequencing technology. For this purpose, differential expression analysis of transcriptome was performed in the liver tissues collected from sheep with higher and lower USFA in their longissimus muscle. In addition, gene polymorphism and association analyses were also performed for the putative candidate genes. Since consumers intake FA from muscle tissues, the longissimus dorsi muscle tissues were used for FA

composition analysis; whereas FA are metabolised in the liver so hepatic transcriptome analysis was performed to unravel the genes and networks controlling FA metabolism in sheep.

# Result

## Phenotypic variation between groups

Phenotypic profile shows the descriptive statistics for fatty acids (FA) composition in Indonesian Javanese fat-tailed sheep (Table 1). Twenty-nine different molecules from FA compositions including total SFA, PUSFA and MUSFA were detected in each of the samples. Total SFA contained thirteen FA, namely capric acid (C10:0), lauric acid (C12:0), tridecan acid (C13:0), myristic acid (C14:0), pentadecanoic acid (C15:0), palmitic acid (C16:0), heptadecanoic acid (C17:0), stearic acid (C18:0), arachidic acid (C20:0), heneicosanoic acid (C21:0), behenic acid (C22:0), tricosanoic acid (C23:0), tetracosanoic acid (C24:0), with an average level of 0.23, 0.47, 0.01, 3.05, 0.51, 18.44, 0.90, 15.78, 0.13, 0.02, 0.06, 0.03, and 0.05%, respectively. Total MUSFA (C14:1; C16:1; C17:1, C18:1n9c, C18:1n9t; C20:1, and C24:1) and PUSFA (C18:2n6c; C18:3n6; C18:3n3, C20:2; C20:3n6, C20:4n6; C22:2, C20:5n3, C22:6n3) were calculated by adding each of the seven and nine FA, respectively. The results also indicated that total SFA was higher than MUSFA and PUSFA (Table 1). The descriptive statistics and the analysis of variance for the FA concentration (expressed in % FA) for higher and lower FA-groups are described in Table 1. There were significant differences ($p < 0.01$) between the higher- and lower-groups of sheep for the concentrations of FA measured in this study (Table 1).

## Quality control and analysis of RNA deep sequencing data

From the sheep (n = 100) population, liver tissues with higher (n = 3) and lower (n = 3) unsaturated fatty acids (USFA) content were selected for high-throughput sequencing. cDNA libraries from 6 samples of sheep liver tissues (3 from HUSFA = higher USFA, and 3 from LUSFA = lower USFA) were sequenced using Illumina HiSeq 2500. The sequencing produced clusters of sequence reads with maximum of 100 base-pair (bp). After quality control and filtering, the total number of reads for liver samples were ranged from 21.28 to 28.51 million with a median of 23.90 million. Total number of reads for each group of samples and the number of reads mapped to reference sequences are shown in Table 2. In case of LUSFA group, 84.51 to 85.69% of total reads were aligned to the reference sequence, whereas 85.20 to 87.38% of the total reads were aligned in case of the HUSFA group.

## Differential gene expression analysis

Differential gene expression from livers tissues of sheep with HUSFA and LUSFA levels were calculated from the raw reads using the R package DESeq. The significance scores were corrected for multiple testing using Benjamini-Hochberg correction. A negative binomial distribution-based method implemented in DESeq was used to identify differentially expressed genes (DEGs) in the liver tissues collected from sheep with divergent unsaturated fatty acids (USFA) level in the longissimus muscle. A total of 198 DEGs were selected from the differential expression analysis using criteria $p$ adjusted $< 0.05$ and log2 fold change $> 1.5$ (Fig 1). In liver tissues, 110 genes were found to be highly expressed in HUSFA group, whereas 98 genes were found to be highly expressed in LUSFA group (S1 Table). The range of log2 fold change values for DEGs were between 4.09 to—4.80 (Fig 2 and Table 3). Heatmaps illustrated the top 30 up- and down-regulated genes identified in the liver tissues from HUSFA and LUSFA sheep. The top 30 up- and down-regulated genes identified in the liver tissues with divergent USFA levels

**Table 1. Descriptive statistic fatty acid composition in Indonesian Javanese fat tailed.**

| Traits | Mean | SD | Lower (n = 3) | | Higher (n = 3) | |
|---|---|---|---|---|---|---|
| | (n = 100) | (n = 100) | Mean | SD | Mean | SD |
| Fat content | 3.66 | 3.24 | 2.91 | 3.45 | 1.18 | 0.54 |
| Capric acid (C10:0) | 0.23 | 1.39 | 0.01 | 0.01 | 0.10 | 0.07 |
| Lauric acid (C12:0) | 0.47 | 0.48 | 0.16 | 0.08 | 0.68 | 0.51 |
| Tridecanoic acid (C13:0) | 0.01 | 0.01 | 0.00 | 0.00 | 0.01 | 0.01 |
| Myristic acid (C14:0) | 3.05 | 1.70 | 0.75[b] | 0.29 | 3.39[a] | 0.55 |
| Myristoleic acid (C14:1) | 0.14 | 0.10 | 0.18 | 0.05 | 0.07 | 0.04 |
| Pentadecanoic acid (C15:0) | 0.51 | 0.17 | 0.26 | 0.06 | 0.47 | 0.24 |
| Palmitic acid (C16:0) | 18.44 | 4.47 | 8.38[b] | 0.90 | 24.30[a] | 2.69 |
| Palmitoleic acid (C16:1) | 1.54 | 0.44 | 0.81 | 0.21 | 1.62 | 0.54 |
| Heptadecanoic acid (C17:0) | 0.90 | 0.33 | 0.52 | 0.05 | 0.69 | 0.39 |
| Ginkgoleic acid (C17:1) | 0.33 | 0.35 | 0.57[a] | 0.15 | 0.03[b] | 0.05 |
| Stearic acid (C18:0) | 15.78 | 5.62 | 12.82 | 1.15 | 14.67 | 7.98 |
| Elaidic acid (C18:1n9t) | 2.91 | 7.16 | 0.01 | 0.00 | 0.01 | 0.00 |
| Oleic acid (C18:1n9c) | 24.52 | 9.53 | 14.24[b] | 1.37 | 34.23[a] | 2.69 |
| Linoleic acid (C18:2n6c) | 2.36 | 1.87 | 4.41 | 0.33 | 6.97 | 8.04 |
| Arachidic acid (C20:0) | 0.13 | 0.10 | 0.30 | 0.05 | 0.25 | 0.04 |
| Cis-11-Eicosenoic acid (C20:1) | 0.02 | 0.08 | 0.26[a] | 0.03 | 0.02[b] | 0.04 |
| Linoleic acid (C18:3n6) | 0.05 | 0.08 | 0.13 | 0.16 | 0.23 | 0.09 |
| Linolenic acid (C18:3n3) | 0.35 | 0.28 | 0.19[b] | 0.06 | 0.67[a] | 0.07 |
| Henecosanoic acid (C21:0) | 0.02 | 0.08 | 0.26 | 0.03 | 0.02 | 0.04 |
| Eicosedienoic acid (C20:2) | 0.05 | 0.05 | 0.04 | 0.02 | 0.22 | 0.26 |
| Behenic acid (C22:0) | 0.06 | 0.09 | 0.26[a] | 0.05 | 0.06[b] | 0.05 |
| Homo-γ linolenic acid (C20:3n6) | 0.07 | 0.13 | 0.33 | 0.12 | 0.26 | 0.46 |
| Arachidonic acid (C20:4n6) | 0.91 | 1.31 | 4.09[a] | 0.36 | 0.83[b] | 0.23 |
| Tricosanoic acid (C23:0) | 0.03 | 0.05 | 0.16[a] | 0.04 | 0.01[b] | 0.02 |
| Tetracosanoic (C24:0) | 0.05 | 0.09 | 0.25[a] | 0.08 | 0.04[b] | 0.07 |
| Eicosapentanoic acid (C20:5n3) | 0.20 | 0.21 | 0.56 | 0.34 | 0.34 | 0.06 |
| Nervonoic acid (C24:1) | 0.04 | 0.09 | 0.17 | 0.11 | 0.07 | 0.01 |
| Cis-4, 7, 10, 13, 16, 19-Docosahexaaonic (C22:6n3) | 0.05 | 0.07 | 0.10 | 0.05 | 0.20 | 0.35 |
| Saturated Fatty Acid (SFA) (%) | 39.73 | 9.22 | 23.92[b] | 2.69 | 44.69[a] | 4.75 |
| Monounsaturated Fatty Acid (MUSFA) (%) | 26.58 | 9.81 | 15.98[b] | 1.62 | 35.96[a] | 2.17 |
| Polyunsaturated Fatty Acid (PUSFA) (%) | 4.02 | 2.84 | 9.86 | 0.87 | 9.62 | 9.05 |
| Unsaturated Fatty Acid (USFA) (%) | 30.60 | 10.12 | 25.84[b] | 2.35 | 45.59[a] | 11.22 |
| Fatty Acid Total (%) | 73.17 | 13.71 | 50.03[b] | 4.89 | 92.53[a] | 4.58 |

Mean ± SD are units of percentage fatty acid composition.

[ab] Mean value with different superscript letters in the same row differ significantly at $P<0.05$.

along with log FC and p values are listed in the Table 3. The differential expression analysis of data revealed both novel transcripts and common genes which were previously identified in various gene expression studies related to FA. Novel transcripts from this analysis and commonly found genes are mentioned in detailed in the discussion section.

## Biological function analysis for DEGs

Gene ontology (GO) and pathway enrichment analysis were performed to gain insight into the predicted genes networks. The most significant GO terms were categorized into biological

**Table 2. Summary of sequence read alignments to reference genome in liver samples.**

| Group | Sample | Total number of reads (million) | Un-mapped reads (million) | Mapped reads (million) | Percentage of unmapped reads | Percentage of mapped reads |
|---|---|---|---|---|---|---|
| Lower unsaturated fatty acid | LUSFA1 | 23.53 | 3.65 | 19.89 | 15.49 | **84.51** |
| | LUSFA2 | 22.36 | 3.28 | 19.08 | 14.67 | **85.33** |
| | LUSFA3 | 28.51 | 4.08 | 24.43 | 14.31 | **85.69** |
| Higher unsaturated fatty acid | HUSFA1 | 22.35 | 2.82 | 19.53 | 12.62 | **87.38** |
| | HUSFA2 | 25.38 | 3.24 | 22.14 | 12.77 | **87.23** |
| | HUSFA3 | 21.28 | 3.17 | 18.22 | 14.80 | **85.20** |

processes, cellular components, and molecular functions (Fig 3). The enriched biological processes identified were mainly related to cytokinesis, glycoprotein metabolic process, mitotic spindle, N linked glycosylation, acute inflammatory response, and regulation of developmental

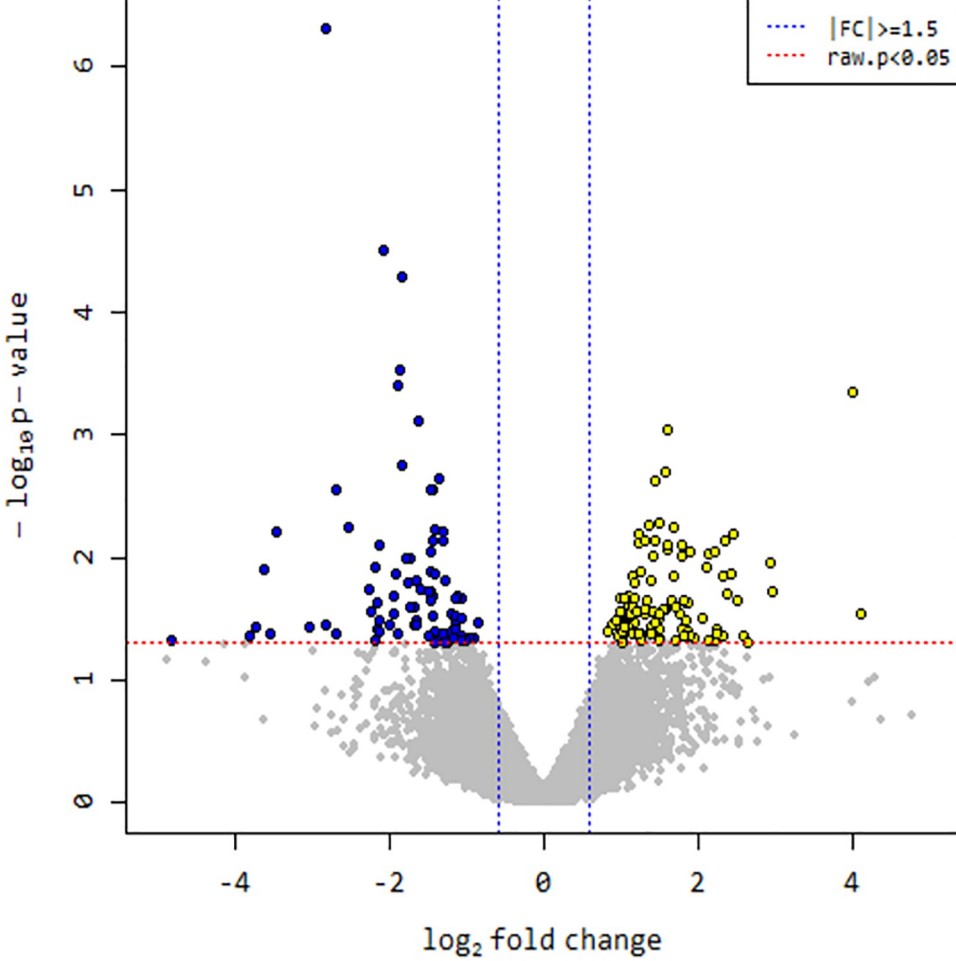

**Fig 1. Volcano plot of the 136 differentially-expressed protein-coding genes.**

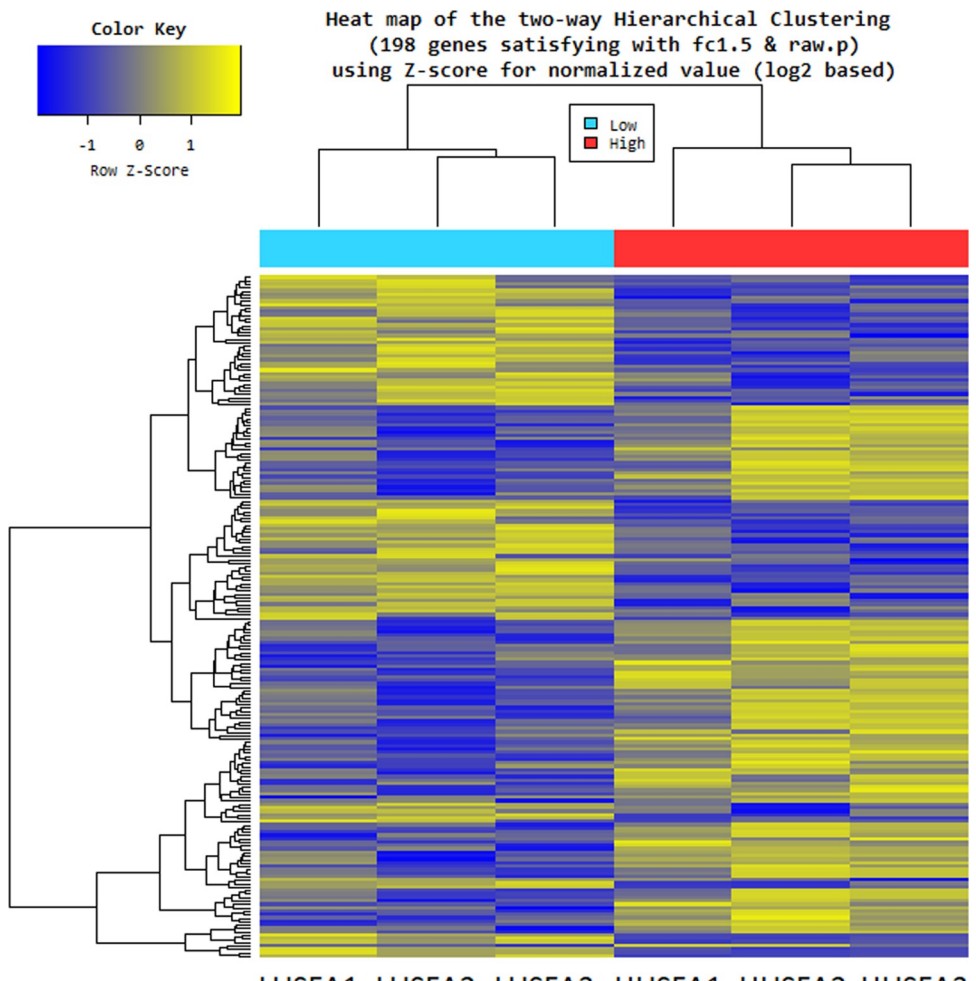

**Fig 2. Heatmap showing differentially expressed genes in liver tissues.**

process. Cellular components consisted of cell projection part, extracellular space, integral to plasma membrane, and proteinaceous extracellular matrix were significantly enriched. The molecular functions identified were related to kinase inhibitor activity, growth factor binding, and GTPase activity. A total of 11 significantly enriched KEGG pathways were identified as overrepresented for the DEGs. The KEGG pathway analysis showed that glycosaminoglycan biosynthesis-keratan sulphate, adipokine signaling, galactose metabolism, endocrine and other factor-regulated calcium metabolism, mineral metabolism, and PPAR signaling pathways were significantly involved in fatty acids metabolism regulation in the liver (Fig 4).

## Regulatory hub genes of the hepatic transcriptome network

In order to identify the key regulatory genes in the transcriptional network, a liver-specific protein-protein interaction (PPI) network was created that comprised of 48 seed genes; and 530 nodes connected with 578 edges. Based on the network centrality measures, the potential Hub genes were identified, among which SOCS3, CBX6, MCM4, ITGB3, TGFBR2, GPRASP1, CELSR3, SDC3, SPOCK1, SEL1L and LEPR were upregulated, whereas ACTA2, GPRASP1, TPM2, TGM3, PTK6, and LTF were downregulated (Fig 5A and 5B). In addition, we have also

**Table 3. Top 30 up- and down-regulated genes in liver tissues collected from sheep with higher and lower unsaturated fatty acids.**

| Gene | Orthologue gene description | Reference ID | Log 2 Fold Change[¥] | p-adj. |
|---|---|---|---|---|
| LOC105607569 | zinc finger protein 549-like | XP_012045546.1 | 4.092012 | 0.03 |
| LOC105606890 | uncharacterized LOC105606890 | | 3.979725 | 0.00 |
| LOC106991076 | | | 2.964076 | 0.02 |
| LOC101113831 | complement C3-like | XP_004022911.2 | 2.930475 | 0.01 |
| LOC101117231 | sialic acid-binding Ig-like lectin 14 | XP_014960758.1 | 2.633583 | 0.05 |
| LOC105608569 | | | 2.593518 | 0.04 |
| LOC105603929 | | | 2.503738 | 0.02 |
| EDAR | ectodysplasin A receptor | XP_014949857.1 | 2.451080 | 0.01 |
| LOC105611460 | uncharacterized LOC105611460 | | 2.416814 | 0.01 |
| CLEC4E | C-type lectin domain family 4 member E | XP_004007622.1 | 2.379035 | 0.02 |
| LOC105605927 | complement C3-like | XP_011963503.1 | 2.333445 | 0.01 |
| SDC3 | syndecan 3 | XP_004005098.1 | 2.317781 | 0.04 |
| LOC101111946 | complement C3-like | XP_004022959.2 | 2.312078 | 0.01 |
| TRNAC-ACA | tRNA-Cys | | 2.245158 | 0.04 |
| CBX6 | chromobox 6 | XP_014949479.1 | 2.244278 | 0.04 |
| TRNAG-GCC | tRNA-Gly | | 2.222962 | 0.05 |
| LOC101111058 | butyrophilin-like protein 1 | XP_004018962.1 | 2.213749 | 0.01 |
| LOC101114799 | low quality protein: tyrosine-protein phosphatase non-receptor type substrate 1-like | XP_012044186.1 | 2.119086 | 0.01 |
| SAMD14 | low quality protein: sterile alpha motif domain-containing protein 14 | XP_014954204.1 | 2.118413 | 0.05 |
| TBC1D30 | TBC1 domain family member 30 | XP_004006543.1 | 1.890920 | 0.01 |
| KBTBD11 | kelch repeat and BTB (POZ) domain containing 11 | XP_014960001.1 | 1.880925 | 0.04 |
| SLC26A6 | solute carrier family 26 (anion exchanger), member 6 | XP_011955481.1 | 1.791577 | 0.01 |
| APOA5 | apolipoprotein A-V | XP_014956330.1 | 1.592786 | 0.01 |
| TGFBR2 | TGF-beta receptor type-2 | XP_011954697.1 | 1.426411 | 0.03 |
| SLC43A2 | large neutral amino acids transporter small subunit 4 | XP_014954050.1 | 1.378811 | 0.04 |
| SLC25A30 | kidney mitochondrial carrier protein 1 | XP_012039782.1 | 1.347998 | 0.01 |
| LEPR | leptin receptor | NP_001009763.1 | 1.155613 | 0.01 |
| GFPT1 | glutamine—fructose-6-phosphate aminotransferase [isomerizing] 1 | XP_014949778.1 | 1.080227 | 0.03 |
| COL27A1 | collagen, type XXVII, alpha 1 | XP_014948447.1 | 1.048113 | 0.04 |
| SLC8A1 | sodium/calcium exchanger 1 | XP_012028463.1 | 1.027025 | 0.04 |
| FAM162B | family with sequence similarity 162 member B | XP_004011224.2 | -1.402540 | 0.04 |
| MESP2 | low quality protein: mesoderm posterior protein 2 | XP_014957268.1 | -1.404600 | 0.01 |
| MYCBPAP | MYCBP associated protein | XP_012041276.1 | -1.413107 | 0.01 |
| LOC105607855 | uncharacterized LOC105607855 | | -1.416474 | 0.05 |
| NAV3 | neuron navigator 3 | XP_004006259.1 | -1.421068 | 0.00 |
| GPRASP1 | G protein-coupled receptor associated sorting protein 1 | XP_014960615.1 | -1.426950 | 0.01 |
| PTK6 | protein tyrosine kinase 6 | XP_004014457.1 | -1.434878 | 0.01 |
| CYP17A1 | cytochrome P450, family 17, subfamily A, polypeptide 1 | NP_001009483.1 | -1.438451 | 0.03 |
| SLC39A10 | solute carrier family 39 (zinc transporter), member 10 | XP_004004828.1 | -1.615897 | 0.00 |
| GSTCD | glutathione S-transferase, C-terminal domain containing | XP_012034961.1 | -1.865130 | 0.00 |
| FABP7 | fatty acid binding protein 7, brain | XP_004011201.1 | -2.125140 | 0.01 |
| LOC101110035 | 40S ribosomal protein S27-like | XP_012001488.1 | -2.129269 | 0.04 |
| LOC105612497 | uncharacterized LOC105612497 | | -2.140696 | 0.04 |
| LOC105604437 | uncharacterized LOC105604437 | | -2.156416 | 0.02 |
| LOC101119043 | zinc finger protein 554 | XP_004008671.1 | -2.190824 | 0.05 |
| NOV | nephroblastoma overexpressed | XP_004011814.2 | -2.191495 | 0.01 |
| LOC106991630 | | | -2.223037 | 0.03 |

*(Continued)*

**Table 3.** (Continued）

| Gene | Orthologue gene description | Reference ID | Log 2 Fold Change[¥] | p-adj. |
|---|---|---|---|---|
| LOC106990988 | | | -2.248084 | 0.02 |
| TMEM253 | Low quality protein: transmembrane protein 253 | XP_014952384.1 | -2.515845 | 0.01 |
| GUCA2A | guanylate cyclase activator 2A (guanylin) | NP_001098731.1 | -2.679505 | 0.04 |
| LTF | Lactotransferrin | NP_001020033.1 | -2.690483 | 0.00 |
| LOC101108292 | guanylate-binding protein 2-like | XP_004022862.2 | -2.823945 | 0.04 |
| CD22 | B-cell receptor CD22 | XP_012045564.1 | -2.827219 | 0.00 |
| DIRAS3 | DIRAS family, GTP-binding RAS-like 3 | NP_001120754.1 | -3.016670 | 0.04 |
| LOC105604312 | uncharacterized LOC105604312 | | -3.445525 | 0.01 |
| LOC101118931 | PWWP domain-containing protein MUM1L1 | XP_011963044.1 | -3.535861 | 0.04 |
| AURKC | aurora kinase C | XP_004015492.1 | -3.611742 | 0.01 |
| PGPEP1L | pyroglutamyl-peptidase 1-like protein | XP_014957351.1 | -3.718097 | 0.04 |
| LOC101114032 | uncharacterized LOC101114032 | | -3.812245 | 0.04 |
| LOC101109629 | olfactory receptor-like protein DTMT | XP_004013311.1 | -4.805143 | 0.05 |

[¥] Positive values of Log2 fold change indicate up regulation and negative values indicate down regulation.

created a liver-specific gene co-expression network to pick up more potential Hub genes, those could have been missed in the PPI network. The co-expression network illustrated that RAC-GAP1, MCM4, SDC3, CKAP2, RNASE6, PREX1, QSOX1, and FUT11 were the upregulated, whereas CDC42EP5, SSC5D, GPRASP1, HRC, NRN1 and TPM2 were the downregulated Hub genes (Fig 6A and 6B). Notably, RACGAP1, TGFBR2, LEPR, MCM4, SDC3, GPRASP1 were the common Hub genes in both PPI and co-expression network analysis (S2 and S3 Tables).

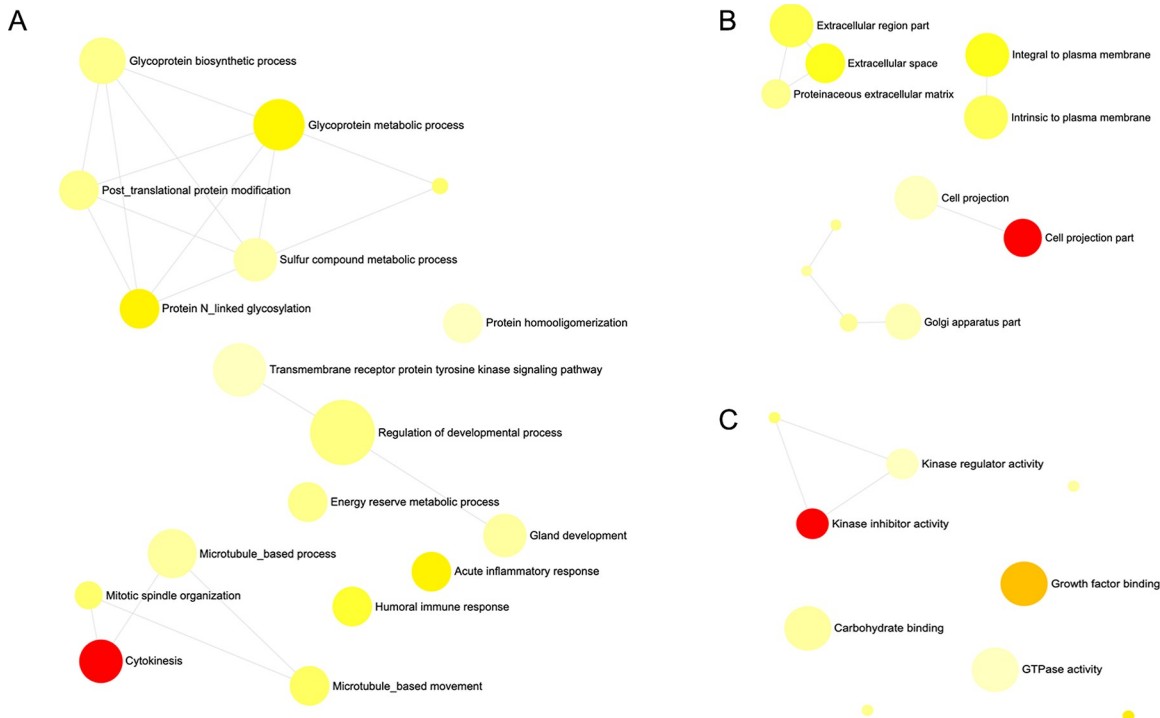

**Fig 3. Network illustration of GO term enrichment classification in Javanese fat–tailed sheep.**

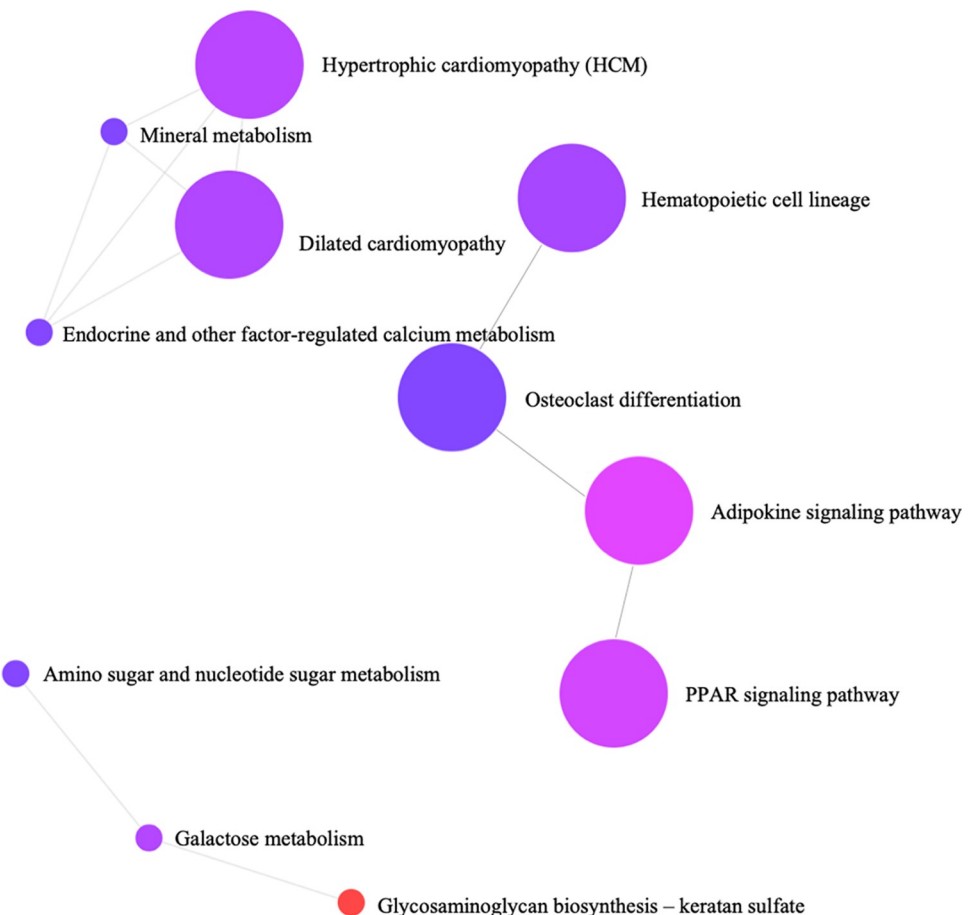

**Fig 4. Network illustration of KEGG pathways in Javanese fat–tailed sheep.**

## Validation of selected DEGs using quantitative Real Time PCR (qRT-PCR)

A total of 8 differentially expressed genes (CYP17A1, FABP7, GSTCD, SLC25A30, APOA5, GFPT1, LEPR and TGFBR2) were selected and quantified using qRT-PCR, as part of RNA-Seq results validation. For this purpose, the same samples used in the RNA-deep sequencing were used. Comparison of qRT-PCR data for 8 selected genes showed quantitative concordance of expression with the RNA-Seq results (Fig 7). Gene expression values for qRT-PCR were normalized using the average expression values of housekeeping gene GAPDH and β-Actin. Details of GenBank accession numbers, primers sequences, product size, and annealing temperature for qRT-PCR validation used in this study are listed in Table 4.

## Gene variation analysis and association study

A total of 226 single nucleotide polymorphisms (SNPs) were identified in 31 DEGs between higher and lower USFA groups (S4 Table). The selected polymorphisms identified in DEGs for liver samples are given in Table 5. The distribution of the number of genes having SNPs, and selected SNPs used for validation are shown in Fig 8A and 8B, respectively. Validation of the SNP results for the association study was carried out by selecting a total of 4 SNPs based on the functional SNPs and the function related to fatty acid metabolism (Fig 8B and S5 Table). The selected SNPs were harboured in APOA5, CFHR5, TGFBR2 and LEPR genes. These SNPs

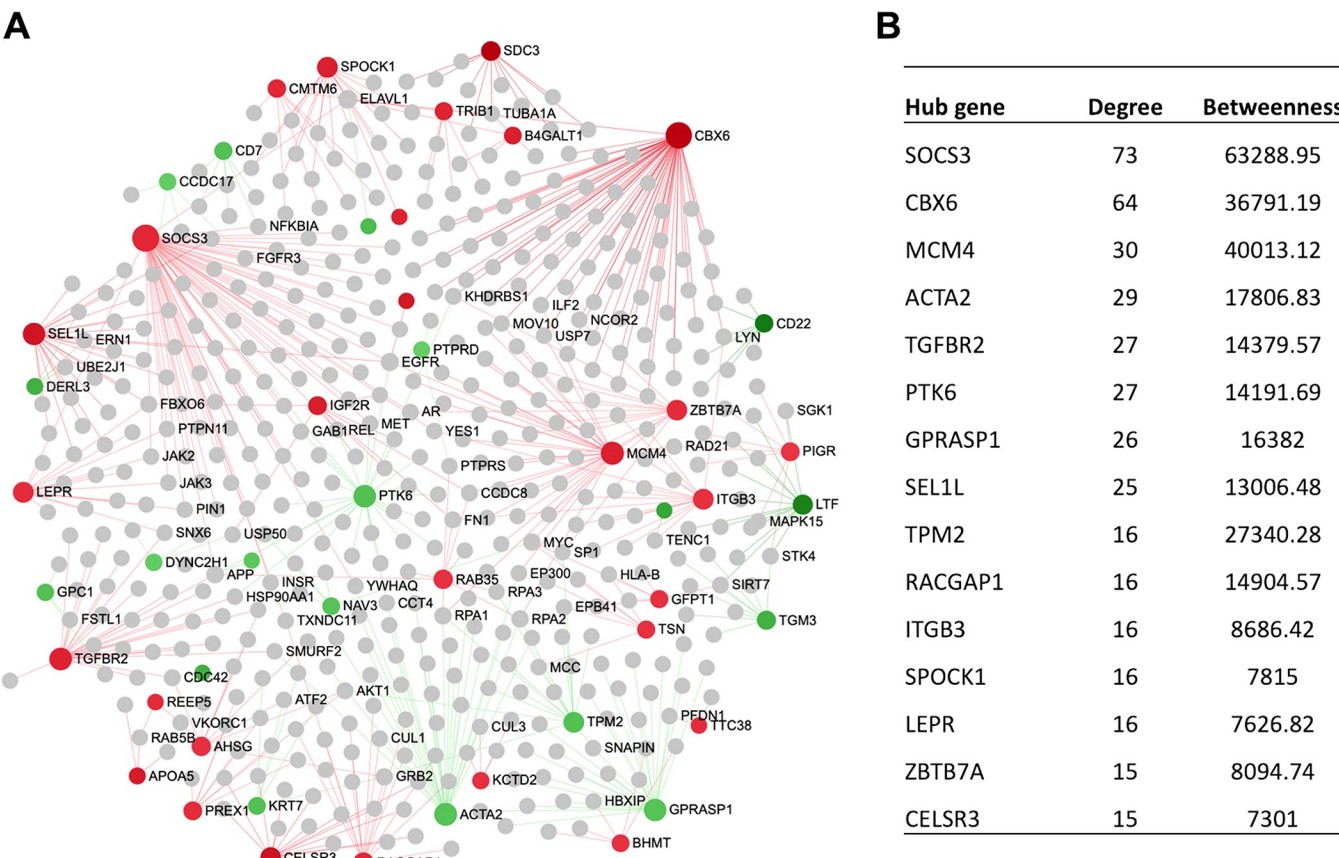

**Fig 5. The liver-specific PPI network generated from the DEGs.**

| Hub gene | Degree | Betweenness |
|---|---|---|
| SOCS3 | 73 | 63288.95 |
| CBX6 | 64 | 36791.19 |
| MCM4 | 30 | 40013.12 |
| ACTA2 | 29 | 17806.83 |
| TGFBR2 | 27 | 14379.57 |
| PTK6 | 27 | 14191.69 |
| GPRASP1 | 26 | 16382 |
| SEL1L | 25 | 13006.48 |
| TPM2 | 16 | 27340.28 |
| RACGAP1 | 16 | 14904.57 |
| ITGB3 | 16 | 8686.42 |
| SPOCK1 | 16 | 7815 |
| LEPR | 16 | 7626.82 |
| ZBTB7A | 15 | 8094.74 |
| CELSR3 | 15 | 7301 |

were analysed to validate their segregation and association in the studied sheep population (n = 100). Our association analyses suggested that, the polymorphisms in APOA5, CFHR5, TGFBR2 and LEPR were associated with fatty acid composition (Table 6) in the studied sheep population.

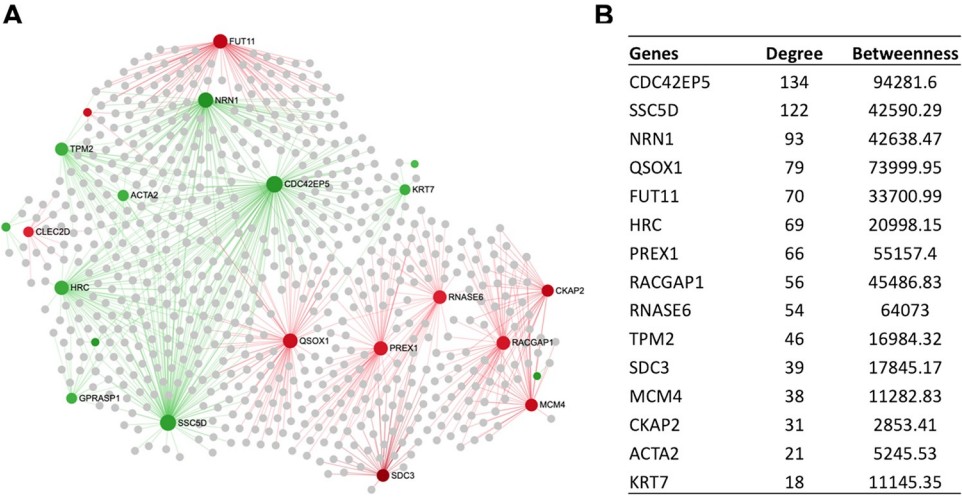

| Genes | Degree | Betweenness |
|---|---|---|
| CDC42EP5 | 134 | 94281.6 |
| SSC5D | 122 | 42590.29 |
| NRN1 | 93 | 42638.47 |
| QSOX1 | 79 | 73999.95 |
| FUT11 | 70 | 33700.99 |
| HRC | 69 | 20998.15 |
| PREX1 | 66 | 55157.4 |
| RACGAP1 | 56 | 45486.83 |
| RNASE6 | 54 | 64073 |
| TPM2 | 46 | 16984.32 |
| SDC3 | 39 | 17845.17 |
| MCM4 | 38 | 11282.83 |
| CKAP2 | 31 | 2853.41 |
| ACTA2 | 21 | 5245.53 |
| KRT7 | 18 | 11145.35 |

**Fig 6. The liver-specific gene co-expression network generated from the DEGs.**

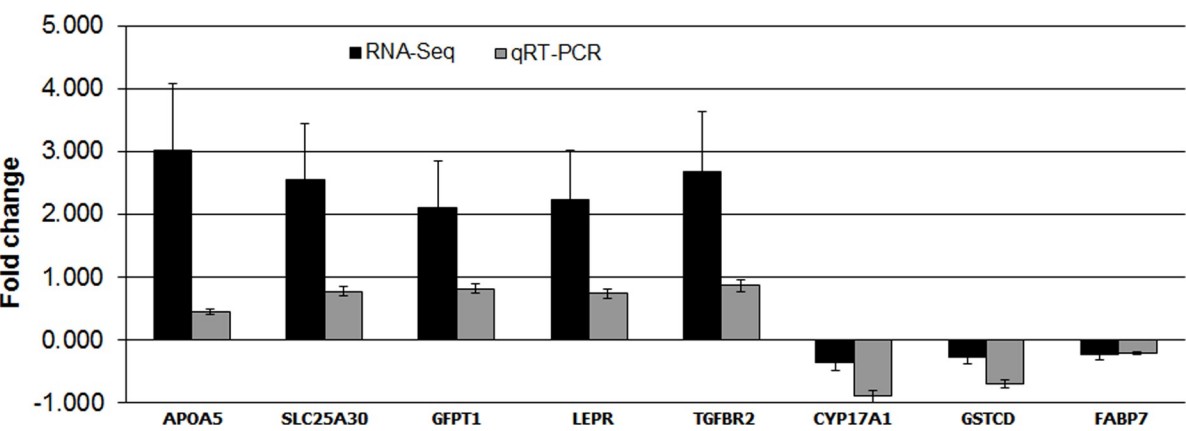

**Fig 7. The qRT-PCR validation.**

**Table 4. GenBank accession numbers and primer sequences for qRT-PCR and genotyping.**

| Gene name | Accession number | Primer sequence | Application | Enzymes | Tm (°C) | Size (bp) | Cutting Size (bp) |
|---|---|---|---|---|---|---|---|
| APOA5 | XM_015100844.1 | F: 5'- GTC ATC CCT CTT TGA ACC TC -3' | qRT-PCR | - | 60 | 208 | - |
| | | R: 5'- CAA GAG GAG GTC CTT AGT TC -3 | | | | | |
| CYP17A1 | NM_001009483.1 | F: 5'- CAC TCT AGA CAT CCT GTC AG-3' | qRT-PCR | - | 60 | 241 | - |
| | | R: 5'- GCT GAT TAT GTT GGT GAC CG -3 | | | | | |
| FABP7 | XM_004011152.3 | F: 5'- CTT TCT GTG CTA CCT GGA AG -3' | qRT-PCR | - | 60 | 267 | - |
| | | R: 5'- CAA GTT TGT CTC CAT CCA GG -3 | | | | | |
| GFPT1 | XM_015094292.1 | F: 5'- GAC TGG AGT ACA GAG GAT AC -3' | qRT-PCR | - | 60 | 203 | - |
| | | R: 5'- CCA ACG GGT ATG AGC TAT TC -3 | | | | | |
| GSTCD | XM_012179572.2 | F: 5'- CGC TTG ACG TTC TTT CTC TC -3' | qRT-PCR | - | 60 | 258 | - |
| | | R: 5'- CTC TTG GCA CTT CCT GAA TC -3 | | | | | |
| LEPR | NM_001009763.1 | F: 5'- GAA GCC TGA TCC ACC ATT AG-3' | qRT-PCR | - | 60 | 239 | - |
| | | R: 5'- CAT CCA ATC TCT TGC TCC TC-3' | | | | | |
| SLC25A30 | XM_012184392.2 | F: 5'- GCT ATG CTT CTG TGA ACG AC-3' | qRT-PCR | - | 60 | 212 | - |
| | | R: 5'- CTA TTC TCA CCA ATG CGT GC-3' | | | | | |
| TGFBR2 | AY751461.1 | F: 5'- CAG ACA TCA ACC TCA AGC AC-3' | qRT-PCR | - | 60 | 281 | - |
| | | R: 5'- CTT GAC CAG GAT GTT GGA GC-3' | | | | | |
| GAPDH | NC_019460.2 | F: 5'- GAGAAACCTGCCAAGTATGA -3' | qRT-PCR | - | 62 | 203 | - |
| | | R: 5'- TACCAGGAAATGAGCTTGAC-3 | | | | | |
| β-Actin | NC_019471.2 | F: 5'- GAAAACGAGATGAGATTGGC -3' | qRT-PCR | - | 62 | 194 | - |
| | | R: 5'- CCATCATAGAGTGGAGTTCG-3 | | | | | |
| LEPR | NC_019458.2 | F: 5'- GAT GAC CTG ACA TAT CCA GG -3' | Genotyping | *AciI* | 60 | 432 | AA: 292 and 140 |
| | | R: 5'- CAA TGA AGT GGG GAA AGG AC -3' | | | | | CC: 432 |
| TGFBR2 | NC_019476.2 | F: 5'-CAG AGA TAA GGC AGT TTG GC-3' | Genotyping | *TaqI* | 55 | 488 | GG: 303, 153 and 32 |
| | | R: 5'-GCA AAA GTA CTC AGG ACA GC-3' | | | | | AA: 456 and 32 |
| APOA5 | NC_019472.2 | F: 5'- CTG CAC AGG ATA GCT GAA GC-3' | Genotyping | *BssSI* | 60 | 258 | CC: 159 and 99 |
| | | R: 5'- CTT TAT CCC AGG GTC TGG TC-3' | | | | | TT: 258 |
| CFHR5 | NC_019469.2 | F: 5'-CTT TCC CAG TTT CTC TTG GG-3' | Genotyping | *AciI* | 60 | 406 | CC: 306 and 100 |
| | | R: 5'-GAC CAG GCT GAT AAC AAA TG-3' | | | | | TT: 406 |

**Table 5. Polymorphisms detected in the highly polymorphic DEGs.**

| Refseq ID | Gene name | Chr | Position | db SNP | Ref | Alt | Higher fatty acid coverage | Higher fatty acid mean phred score | Lower fatty acid coverage | Lower fatty acid mean phred score | Sample group | SNP clasification |
|---|---|---|---|---|---|---|---|---|---|---|---|---|
| XM_027979224.1 | APOA5 | 15 | 26896190 | . | T | C | 2242,5 | 225 | 0 | 0 | Higher | Downstream gene variant |
| XM_027979224.1 | APOA5 | 15 | 26896453 | . | GA | GAA | 180 | 228 | 0 | 0 | Higher | 3 prime UTR variant |
| XM_027979224.1 | APOA5 | 15 | 26896677 | rs402578508 | C | T | 2253,333333 | 228 | 943,3333333 | 226 | Higher and Lower | 3 prime UTR variant |
| XM_027979224.1 | APOA5 | 15 | 26896823 | . | C | A | 0 | 0 | 831 | 222 | Lower | Missense variant |
| XM_027979224.1 | APOA5 | 15 | 26897295 | rs589107798 | A | G | 588,6666667 | 228 | 179 | 228 | Higher and Lower | Synonymous variant |
| XM_027979224.1 | APOA5 | 15 | 26897513 | . | G | C | 108 | 103 | 0 | 0 | Higher | Missense variant |
| XM_027979224.1 | APOA5 | 15 | 26897515 | . | A | C | 135 | 76 | 0 | 0 | Higher | Missense variant |
| XM_027976096.1 | CFHR5 | 12 | 74015573 | rs424959076 | C | T | 105 | 222 | 0 | 0 | Higher | Synonymous variant |
| XM_027976096.1 | CFHR5 | 12 | 74022143 | . | C | G | 103 | 221 | 0 | 0 | Higher | Missense variant |
| XM_027976096.1 | CFHR5 | 12 | 74022229 | rs409473546 | A | T | 123 | 222 | 0 | 0 | Higher | Missense variant |
| XM_027976096.1 | CFHR5 | 12 | 74025194 | rs418356059 | G | T | 228 | 228 | 0 | 0 | Higher | Synonymous variant |
| XM_027976096.1 | CFHR5 | 12 | 74025545 | rs398497259 | A | G | 140 | 222 | 0 | 0 | Higher | Missense variant |
| XM_027976096.1 | CFHR5 | 12 | 74025588 | rs413612756 | A | C | 114 | 221 | 0 | 0 | Higher | Synonymous variant |
| XM_027976096.1 | CFHR5 | 12 | 74025612 | rs420952834 | T | G | 116 | 221 | 0 | 0 | Higher | Synonymous variant |
| XM_027976096.1 | CFHR5 | 12 | 74025633 | rs399169608 | C | T | 130 | 222 | 0 | 0 | Higher | Synonymous variant |
| XM_027976096.1 | CFHR5 | 12 | 74029504 | rs412859061 | A | G | 160 | 222 | 0 | 0 | Higher | Missense variant |
| XM_027976096.1 | CFHR5 | 12 | 74029546 | rs424012492 | T | C | 153 | 222 | 0 | 0 | Higher | Missense variant |
| XM_027976096.1 | CFHR5 | 12 | 74037986 | rs422073184 | T | C | 157 | 228 | 0 | 0 | Higher | Missense variant |
| XM_027976096.1 | CFHR5 | 12 | 74038411 | rs421338064 | G | C | 242,6666667 | 226 | 162 | 228 | Higher and Lower | 3 prime UTR variant |
| XM_027976096.1 | CFHR5 | 12 | 74038439 | rs399840874 | A | G | 228,6666667 | 226 | 172 | 228 | Higher and Lower | 3 prime UTR variant |
| XM_027976096.1 | CFHR5 | 12 | 74038482 | rs411172678 | G | A | 135,3333333 | 226 | 0 | 0 | Higher | 3 prime UTR variant |
| XM_027976096.1 | CFHR5 | 12 | 74038562 | . | ATTT | AT | 143 | 125,5 | 0 | 0 | Higher | 3 prime UTR variant |
| XM_027976096.1 | CFHR5 | 12 | 74038610 | rs404884033 | C | T | 210,3333333 | 226 | 117 | 228 | Higher and Lower | 3 prime UTR variant |
| XM_027976096.1 | CFHR5 | 12 | 74038748 | rs422967211 | A | G | 351,6666667 | 226 | 268 | 228 | Higher and Lower | 3 prime UTR variant |
| XM_027976096.1 | CFHR5 | 12 | 74038806 | rs405523237 | G | C | 300,6666667 | 224 | 213,5 | 228 | Higher and Lower | 3 prime UTR variant |

*(Continued)*

**Table 5.** (Continued)

| Refseq ID | Gene name | Chr | Position | db SNP | Ref | Alt | Higher fatty acid coverage | Higher fatty acid mean phred score | Lower fatty acid coverage | Lower fatty acid mean phred score | Sample group | SNP clasification |
|---|---|---|---|---|---|---|---|---|---|---|---|---|
| XM_027976096.1 | CFHR5 | 12 | 74038847 | rs416824949 | T | C | 266,6666667 | 224 | 220 | 228 | Higher and Lower | 3 prime UTR variant |
| XM_027976096.1 | CFHR5 | 12 | 74039043 | rs402360719 | A | G | 0 | 0 | 105 | 228 | Lower | 3 prime UTR variant |
| XM_027966670.1 | GFPT1 | 3 | 38804055 | rs405549722 | T | C | 225 | 222 | 192,5 | 225 | Higher and Lower | Downstream gene variant |
| XM_027966670.1 | GFPT1 | 3 | 38804078 | . | GCC | GC | 142 | 222 | 193 | 225 | Higher and Lower | Downstream gene variant |
| XM_027966670.1 | GFPT1 | 3 | 38804295 | rs428116355 | G | T | 177 | 221 | 103 | 228 | Higher and Lower | Downstream gene variant |
| XM_012179571.3 | GSTCD | 6 | 19457076 | . | A | C | 2099 | 228 | 7097,333333 | 228 | Higher and Lower | Intron variant |
| XM_012179571.3 | GSTCD | 6 | 19457098 | . | T | C | 4279,333333 | 228 | 8015,666667 | 228 | Higher and Lower | Intron variant |
| NM_001009763.1 | LEPR | 1 | 40761672 | rs407713277 | A | C | 111 | 228 | 0 | 0 | Higher | Downstream gene variant |
| NM_001009763.1 | LEPR | 1 | 40763013 | rs416805159 | G | A | 173 | 228 | 152 | 228 | Higher and Lower | Downstream gene variant |
| XM_015098055.2 | SLC25A30 | 10 | 15911186 | rs406979082 | T | C | 101 | 221 | 0 | 0 | Higher | 3 prime UTR variant; Downstream gene variant |
| XM_015098055.2 | SLC25A30 | 10 | 15911187 | rs422179448 | G | A | 102 | 221 | 0 | 0 | Higher | 3 prime UTR variant; Downstream gene variant |
| XM_015098055.2 | SLC25A30 | 10 | 15912281 | rs418887961 | T | G | 103 | 222 | 0 | 0 | Higher | Downstream gene variant |
| XM_015098055.2 | SLC25A30 | 10 | 15912283 | rs401535429 | T | A | 102 | 222 | 0 | 0 | Higher | Downstream gene variant |
| XM_015098055.2 | SLC25A30 | 10 | 15912963 | rs159417115 | G | A | 209 | 222 | 0 | 0 | Higher | Downstream gene variant |
| XM_027957940.1 | TGFBR2 | 19 | 5105529 | rs161225113 | G | A | 0 | 0 | 103 | 228 | Lower | Downstream gene variant |
| XM_027957940.1 | TGFBR2 | 19 | 5105758 | rs193644594 | A | G | 147 | 222 | 186 | 228 | Higher and Lower | Downstream gene variant |

## Discussion

### Analysis of RNA seq data

This study describes the transcriptome profiles of the liver tissues collected from sheep with higher and lower unsaturated fatty acids (HUSFA vs LUSFA) content in their longisimuss muscle. RNA-Seq have allowed for the large-scale analysis of genomic data, providing new

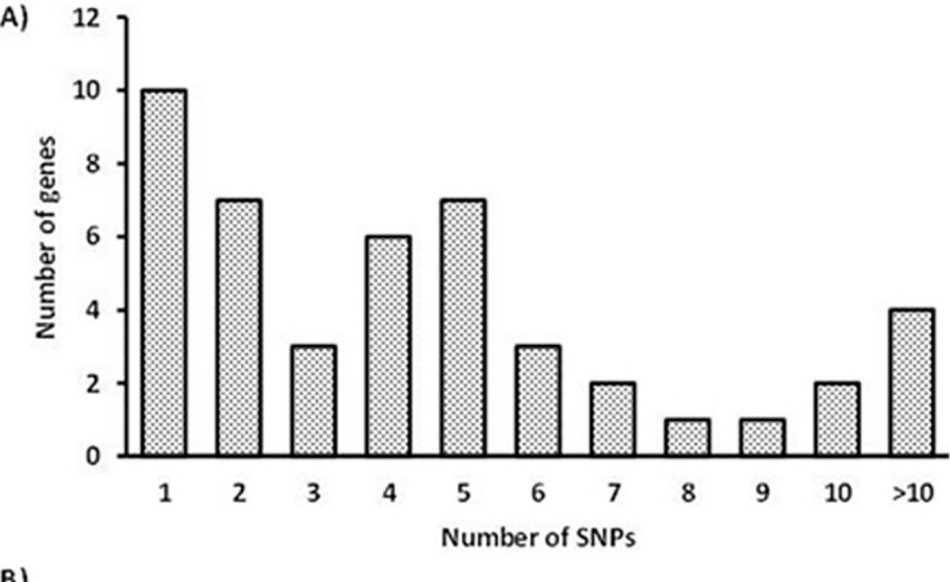

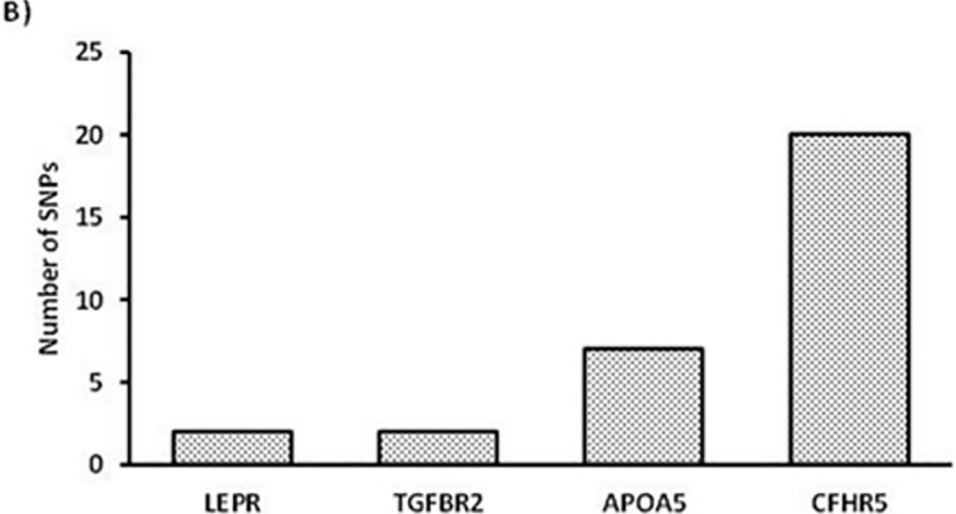

**Fig 8. Distribution of the number of SNPs detected in the DEGs.**

opportunities for the characterization of transcriptome architectures [19]. According to the mapping results, the average number of reads was 23.90 million reads, and on an average 85.89% of the reads were categorized as mapped reads corresponding to exon reads (Table 2). High-quality reads of mapping results were obtained from an RNA-Seq analysis of the six libraries by comparing to the *Ovis aries* genome. The proportion of reads mapped to exons of annotated genes was in accordance (85.70–86.95%) with the previous studies [20–22] in sheep muscle transcriptome, but was higher than that reported by Wang et al. [12] (68.97%) in short-tailed sheep adipose tissue. The percentage of annotated reads varies from 66.40% to 86.95% in sheep transcriptome studies [12, 20–22] supporting our results. The differences between mapping percentages might be due to the current reference transcriptome assembly that might not cover all the transcribed mRNA [23] and consequently low abundant transcripts are less likely to be mapped to the transcriptome assembly [24]. Illumina sequencing data have been described as replicable with relatively little technical variation [25]. Therefore,

**Table 6. Genotypes and association analysis of selected candidate genes in fatty acid composition.**

| Fatty acid composition (%) | APOA5 C>T Genotype (μ±S.D) | | | CFHR5 C>T Genotype (μ±S.D) | | | TGFBR2 A>G Genotype (μ±S.D) | | | LEPR A>C Genotype (μ±S.D) | | |
|---|---|---|---|---|---|---|---|---|---|---|---|---|
| | CC (n = 56) | CT (n = 38) | TT (n = 6) | CC (n = 38) | CT (n = 49) | TT (n = 13) | AA (n = 82) | AG (n = 15) | GG (n = 3) | AA (n = 64) | AC (n = 32) | CC (n = 4) |
| Fat content | 4.68 ± 3.76a | 2.24 ± 1.42b | 3.11 ± 3.16ab | 2.55 ± 2.05b | 3.70 ± 3.13ab | 4.88 ± 4.83a | 3.28 ± 3.13 | 3.56 ± 2.23 | 2.00 ± 1.76 | 4.00 ± 3.53 | 2.95 ± 2.35 | 3.86 ± 4.38 |
| Caprilic acid (C8:0) | 0.00 ± 0.00b | 0.10 ± 0.16a | 0.00 ± 0.00b | 0.03 ± 0.10 | 0.05 ± 0.12 | 0.00 ± 0.00 | 0.04 ± 0.11 | 0.03 ± 0.08 | 0.00 ± 0.00 | 0.02 ± 0.06 | 0.07 ± 0.16 | 0.00 ± 0.00 |
| Capric acid (C10:0) | 0.08 ± 0.05 | 0.47 ± 2.25 | 0.08 ± 0.05 | 0.08 ± 0.06 | 0.37 ± 1.98 | 0.07 ± 0.05 | 0.25 ± 1.53 | 0.09 ± 0.05 | 0.05 ± 0.04 | 0.31 ± 1.73 | 0.08 ± 0.04 | 0.10 ± 0.11 |
| Lauric acid (C12:0) | 0.46 ± 0.43 | 0.49 ± 0.56 | 0.29 ± 0.17 | 0.43 ± 0.39 | 0.46 ± 0.53 | 0.47 ± 0.47 | 0.46 ± 0.47 | 0.39 ± 0.39 | 0.62 ± 0.93 | 0.54 ± 0.54 | 0.33 ± 0.29 | 0.40 ± 0.21 |
| Tridecanoic acid (C13:0) | 0.01 ± 0.01 | 0.01 ± 0.01 | 0.01 ± 0.01 | 0.007 ± 0.01b | 0.01 ± 0.01a | 0.01 ± 0.01a | 0.01 ± 0.01 | 0.01 ± 0.01 | 0.01 ± 0.01 | 0.01 ± 0.01 | 0.01 ± 0.01 | 0.01 ± 0.01 |
| Myristic acid (C14:0) | 3.17 ± 1.73 | 2.91 ± 1.64 | 2.81 ± 1.94 | 2.89 ± 1.71 | 3.02 ± 1.80 | 2.93 ± 1.67 | 3.04 ± 1.80 | 2.44 ± 0.99 | 2.90 ± 3.31 | 3.34 ± 1.80 | 2.41 ± 1.13 | 3.48 ± 2.75 |
| Myristoleic acid (C14:1) | 0.15 ± 0.11 | 0.13 ± 0.07 | 0.11 ± 0.08 | 0.11 ± 0.07b | 0.13 ± 0.07b | 0.21 ± 0.20a | 0.14 ± 0.11 | 0.10 ± 0.05 | 0.11 ± 0.13 | 0.49 ± 0.16 | 0.12 ± 0.10 | 0.11 ± 0.06 |
| Pentadecanoic acid (C15:0) | 0.49 ± 0.15 | 0.53 ± 0.17 | 0.44 ± 0.23 | 0.45 ± 0.19 | 0.51 ± 0.17 | 0.52 ± 0.21 | 0.49 ± 0.18a | 0.50 ± 0.15a | 0.15 ± 0.24b | 0.49 ± 0.16 | 0.54 ± 0.17 | 0.49 ± 0.08 |
| Palmitic acid (C16:0) | 18.32 ± 3.49b | 19.14 ± 5.10a | 15.08 ± 7.11b | 17.21 ± 6.38 | 18.59 ± 4.23 | 17.46 ± 5.97 | 17.78 ± 5.67 | 18.61 ± 3.99 | 13.12 ± 10.64 | 18.72 ± 4.48 | 17.96 ± 4.50 | 17.66 ± 4.83 |
| Palmitoleic acid (C16:1) | 1.53 ± 0.41 | 1.59 ± 0.43 | 1.25 ± 0.69 | 1.35 ± 0.54 | 1.58 ± 0.42 | 1.48 ± 0.59 | 1.47 ± 0.53 | 1.45 ± 0.36 | 1.06 ± 0.94 | 1.58 ± 0.45 | 1.48 ± 0.41 | 1.28 ± 0.52 |
| Heptadecanoic acid (C17:0) | 1.01 ± 0.37a | 0.76 ± 0.18b | 0.76 ± 0.38b | 0.74 ± 0.30b | 0.94 ± 0.34ab | 0.98 ± 0.51a | 0.85 ± 0.35 | 0.89 ± 0.30 | 0.55 ± 0.51 | 0.90 ± 0.33 | 0.91 ± 0.36 | 0.87 ± 0.25 |
| Ginkgolic acid (C17:1) | 0.49 ± 0.34a | 0.09 ± 0.21b | 0.26 ± 0.21b | 0.21 ± 0.25b | 0.32 ± 0.37b | 0.51 ± 0.37a | 0.29 ± 0.33 | 0.28 ± 0.31 | 0.28 ± 0.24 | 0.34 ± 0.34 | 0.31 ± 0.36 | 0.24 ± 0.18 |
| Stearic acid (C18:0) | 15.39 ± 6.43 | 16.80 ± 3.40 | 12.87 ± 8.06 | 14.67 ± 5.81 | 16.20 ± 5.81 | 15.54 ± 7.76 | 15.46 ± 6.31a | 16.74 ± 4.88a | 8.62 ± 7.00b | 15.03 ± 5.56 | 17.09 ± 5.66 | 17.18 ± 5.22 |
| Elaidic acid (C18:1n9t) | 4.31 ± 9.27 | 1.26 ± 1.53 | 0.021 ± 0.43 | 2.45 ± 5.90 | 2.90 ± 7.21 | 4.60 ± 10.10 | 3.48 ± 7.78 | 0.60 ± 1.15 | 0.38 ± 0.66 | 3.47 ± 8.12 | 1.40 ± 3.56 | 5.82 ± 11.54 |
| Oleic acid (C18:1n9c) | 22.89 ± 10.53 | 27.47 ± 6.88 | 20.98 ± 10.85 | 22.60 ± 10.40 | 25.11 ± 9.82 | 21.16 ± 12.22 | 22.77 ± 11.02 | 27.28 ± 5.61 | 18.28 ± 15.83 | 24.26 ± 10.04 | 25.69 ± 8.00 | 19.26 ± 12.90 |
| Linoleic acid (C18:2n6c) | 2.33 ± 2.34 | 2.49 ± 0.95 | 1.87 ± 1.12 | 2.50 ± 2.62 | 2.25 ± 1.20 | 2.02 ± 1.59 | 2.28 ± 2.05 | 2.51 ± 0.83 | 1.85 ± 1.94 | 2.37 ± 2.14 | 2.41 ± 1.17 | 1.86 ± 1.81 |
| Linolelaidic Acid (C18:2n9t) | 0.00 ± 0.00b | 0.08 ± 0.10a | 0.00 ± 0.00b | 0.04 ± 0.08 | 0.02 ± 0.07 | 0.01 ± 0.05 | 0.03 ± 0.07 | 0.03 ± 0.08 | 0.00 ± 0.00 | 0.02 ± 0.06 | 0.04 ± 0.09 | 0.02 ± 0.06 |
| Linolenic acid (C18:3n3) | 0.23 ± 0.20b | 0.54 ± 0.26a | 0.20 ± 0.22b | 0.36 ± 0.26 | 0.36 ± 0.26 | 0.21 ± 0.18 | 0.32 ± 0.27 | 0.44 ± 0.33 | 0.26 ± 0.26 | 0.32 ± 0.07 | 0.41 ± 0.31 | 0.19 ± 0.23 |
| v-Linolenic acid (C18:3n6) | 0.03 ± 0.06 | 0.03 ± 0.06 | 0.02 ± 0.04 | 0.36 ± 0.08 | 0.02 ± 0.04 | 0.01 ± 0.02 | 0.03 ± 0.06 | 0.01 ± 0.01 | 0.006 ± 0.011 | 0.03 ± 0.07 | 0.01 ± 0.02 | 0.02 ± 0.05 |
| Arachidic acid (C20:0) | 0.09 ± 0.09ab | 0.15 ± 0.08a | 0.08 ± 0.06b | 0.12 ± 0.09 | 0.11 ± 0.08 | 0.11 ± 0.12 | 0.11 ± 0.10 | 0.13 ± 0.05 | 0.04 ± 0.03 | 0.11 ± 0.08 | 0.14 ± 0.10 | 0.07 ± 0.11 |
| Eicosenoic acid (C20:1) | 0.01 ± 0.06 | 0.03 ± 0.10 | 0.00 ± 0.00 | 0.02 ± 0.07 | 0.01 ± 0.07 | 0.04 ± 0.08 | 0.02 ± 0.08 | 0.01 ± 0.06 | 0.00 ± 0.00 | 0.02 ± 0.06 | 0.03 ± 0.10 | 0.00 ± 0.00 |
| Eicosedienoic acid (C20:2) | 0.06 ± 0.06 | 0.03 ± 0.02 | 0.03 ± 0.02 | 0.05 ± 0.07 | 0.04 ± 0.02 | 0.05 ± 0.02 | 0.04 ± 0.05 | 0.04 ± 0.01 | 0.04 ± 0.03 | 0.05 ± 0.06 | 0.04 ± 0.02 | 0.05 ± 0.01 |
| Cis-8,11,14-Eicosetrienoic acid (C20:3n6) | 0.07 ± 0.12 | 0.06 ± 0.10 | 0.02 ± 0.04 | 0.08 ± 0.15 | 0.05 ± 0.06 | 0.06 ± 0.09 | 0.07 ± 0.12 | 0.05 ± 0.04 | 0.02 ± 0.04 | 0.07 ± 0.12 | 0.06 ± 0.08 | 0.06 ± 0.07 |
| Arachidonic acid (C20:4n6) | 1.03 ± 1.43 | 0.78 ± 1.21 | 0.55 ± 0.46 | 0.99 ± 1.42 | 0.76 ± 1.03 | 1.18 ± 1.90 | 0.97 ± 1.39 | 0.64 ± 0.87 | 0.38 ± 0.39 | 0.90 ± 1.28 | 0.90 ± 1.42 | 1.07 ± 0.97 |
| Cis-5,8,11,14,17-Eicosapentaenoic acid (C20:5n3) | 0.10 ± 0.16b | 0.34 ± 0.19a | 0.11 ± 0.16b | 0.22 ± 0.21 | 0.18 ± 0.18 | 0.14 ± 0.25 | 0.20 ± 0.21 | 0.19 ± 0.18 | 0.04 ± 0.04 | 0.17 ± 0.18 | 0.24 ± 0.24 | 0.11 ± 0.02 |
| Heneicosylic acid (C21:0) | 0.01 ± 0.02b | 0.03 ± 0.02a | 0.006 ± 0.010b | 0.02 ± 0.02 | 0.02 ± 0.02 | 0.02 ± 0.02 | 0.02 ± 0.02ab | 0.03 ± 0.02a | 0.00 ± 0.00b | 0.02 ± 0.02 | 0.03 ± 0.02 | 0.02 ± 0.02 |
| Behenic acid (C22:0) | 0.06 ± 0.08 | 0.06 ± 0.09 | 0.03 ± 0.02 | 0.06 ± 0.09 | 0.05 ± 0.07 | 0.07 ± 0.09 | 0.06 ± 0.09 | 0.04 ± 0.04 | 0.02 ± 0.01 | 0.05 ± 0.07 | 0.07 ± 0.10 | 0.07 ± 0.06 |
| Erucic acid (C22:1n9) | 0.00 ± 0.00b | 0.002 ± 0.005a | 0.00 ± 0.00b | 0.001 ± 0.004 | 0.0008 ± 0.0027 | 0.00 ± 0.00 | 0.0007 ± 0.002 | 0.002 ± 0.005 | 0.00 ± 0.00 | 0.0007 ± 0.002 | 0.001 ± 0.004 | 0.00 ± 0.00 |
| Cis-13,16-Docosadienoic acid (C22:2) | 0.01 ± 0.04 | 0.00 ± 0.00 | 0.00 ± 0.00 | 0.003± 0.021 | 0.006 ± 0.042 | 0.01 ± 0.05 | 0.007 ± 0.04 | 0.00 ± 0.00 | 0.00 ± 0.00 | 0.009 ± 0.04 | 0.00 ± 0.00 | 0.00 ± 0.00 |
| Docosahexaaonic acid (C22:6n3) | 0.04 ± 0.08 | 0.04 ± 0.03 | 0.06 ± 0.07 | 0.06 ± 0.10 | 0.03 ± 0.03 | 0.03 ± 0.04 | 0.04 ± 0.07 | 0.03 ± 0.02 | 0.04 ± 0.04 | 0.04 ± 0.08 | 0.03 ± 0.04 | 0.08 ± 0.02 |
| Tricosanoic acid (C23:0) | 0.03 ± 0.05 | 0.02 ± 0.05 | 0.01 ± 0.01 | 0.03 ± 0.05 | 0.02 ± 0.04 | 0.04 ± 0.06 | 0.03 ± 0.05 | 0.02 ± 0.03 | 0.01 ± 0.01 | 0.02 ± 0.04 | 0.03 ± 0.06 | 0.03 ± 0.03 |
| Tetracosanoic acid (C24:0) | 0.05 ± 0.09 | 0.04 ± 0.10 | 0.01 ± 0.02 | 0.05 ± 0.10 | 0.03 ± 0.08 | 0.07 ± 0.12 | 0.05 ± 0.10 | 0.02 ± 0.04 | 0.006 ± 0.011 | 0.04 ± 0.08 | 0.05 ± 0.12 | 0.04 ± 0.05 |
| Nervonic acid (C24:1) | 0.04 ± 0.08 | 0.04 ± 0.11 | 0.003 ± 0.008 | 0.03 ± 0.07 | 0.03 ± 0.09 | 0.06 ± 0.10 | 0.04 ± 0.09 | 0.02 ± 0.02 | 0.00 ± 0.00 | 0.03 ± 0.06 | 0.05 ± 0.13 | 0.03 ± 0.03 |
| Fatty acid total | 72.65 ± 8.32a | 76.28 ± 15.91a | 58.23 ± 27.18b | 68.06 ± 21.72 | 74.03 ± 13.48 | 70.20 ± 21.34 | 70.81 ± 19.43a | 73.79 ± 11.46a | 48.99 ± 40.02b | 73.35 ± 13.30 | 73.10 ± 15.25 | 70.72 ± 8.29 |
| Saturated fatty acid (SFA) | 39.24 ± 7.72ab | 41.57 ± 9.66a | 32.51 ± 15.66b | 36.87 ± 12.25 | 40.44 ± 9.41 | 38.34 ± 13.73 | 38.70 ± 11.84a | 40.02 ± 8.06a | 26.13 ± 21.26b | 39.64 ± 9.19 | 39.79 ± 9.88 | 40.47 ± 4.54 |
| Monounsaturated fatty acid (MUFA) | 25.12 ± 10.83 | 29.35 ± 7.11 | 22.61 ± 11.74 | 24.33 ± 10.79 | 27.18 ± 10.12 | 23.43 ± 12.82 | 24.73 ± 11.42 | 29.15 ± 5.91 | 19.73 ± 17.08 | 26.38 ± 10.30 | 27.67 ± 8.31 | 20.93 ± 13.27 |
| Polyunsaturated fatty acid (PUFA) | 3.92 ± 3.26 | 4.33 ± 2.24 | 2.87 ± 1.70 | 4.33 ± 3.64 | 3.72 ± 1.89 | 3.73 ± 3.66 | 3.98 ± 3.10 | 3.94 ± 1.70 | 2.65 ± 2.55 | 3.99 ± 3.00 | 4.13 ± 2.58 | 3.47 ± 2.58 |
| Unsaturated fatty acid (UFA) | 29.04 ± 11.23ab | 33.68 ± 6.82a | 25.49 ± 12.73b | 28.66 ± 11.95 | 30.91 ± 10.20 | 27.17 ± 13.76 | 28.72 ± 12.18 | 33.10 ± 5.20 | 22.39 ± 19.34 | 30.37 ± 10.73 | 31.81 ± 8.38 | 24.40 ± 12.85 |

the findings of this study clearly demonstrated the power of RNA-Seq and provide further insights into the transcriptome of liver tissues at a finer resolution in sheep.

## Differential express gene analysis

A total of 198 genes were differentially regulated in liver tissues from sheep with divergent USFA levels (S1 Table). The top up- and down-regulated genes in the liver tissues were Zinc Finger Protein 549 with log2 fold change 4.09, and olfactory receptor-like protein DTMT with log2 fold change -4.80, respectively (Table 3). The genes encode Zinc-finger proteins are involved in cell proliferation and differentiation [26] as well as regulate lipid metabolism [27]. However, the relation between olfactory receptor family genes and USFA is yet to understand.

Among the DEGs screened with stringent criteria in the present study, a large proportion of key genes involved in FA biosynthesis, fat deposition, adipogenesis, and lipid metabolism were identified, such as APOA5, SLC25A30, GFPT1, LEPR, TGFBR2, FABP7, GSTCD and CYP17A. APOA5 regulates the assembly and secretion of lipoproteins [28] and controls the plasma triglyceride levels in humans and mice [29, 30]. Interestingly four members of SLC family genes were found to be differentially regulated in this study. SLC8A1 and SLC43A2 were found to be up-regulated, whereas SLC39A10 was found to be down-regulated in the HUSFA group (Table 2). Two members of SLC genes (SLC16A7 and SLC27A6) were reported to be involved in FA metabolism [16]. Kaler and Prasad [31] postulated that SLC39A10 plays an essential role in cell proliferation and migration. However, the mechanism of SLC39A10 downregulation in FA metabolism is not yet clear, so further investigations are warranted to elucidate the function of this novel transcript regarding to FA metabolism. Sodhi et al. [32] reported that Glutamine fructose- 6-phosphate transaminase 1 (GFPT1) is involved in glucose metabolism and differentially expressed in adipose tissue. A mutation in the exon of LEPR (p. Leu663Phe) is reported to be associated with increased feed intake and fatness in pigs [33].

Another gene family found to be differentially expressed that includes CYP17A, GSTCD and FABP7. These three genes were found to be down regulated in the higher USFA sheep in this study. Cytochrome P450 17A1 (CYP17A1, 17α-hydroxylase, 17,20-lyase) belongs to the cytochrome P450 super family that is expressed in the adrenals and gonads [34]. CYP2A6 gene is reported to be involved in meat flavour and odour-related molecules metabolism in sheep [35]. Barone et al. [36] reported that overexpression of CYP17A1 mRNA is associaed with enhancement of conjugated linoleic acid (CLA). The CLA refers to a group of positional and geometrical isomers of linoleic acid (cis-9, cis-12-octadecadienoic acid), an omega-6 essential fatty acid, that exhibit various physiological effects including anti-adipogenic, anti-carcino-genic, and immunomodulatory effect [37]. Glutathione S-transferase, C-terminal domain (GSTCD) belongs to the Glutathione S-transferases (GSTs) family that are functionally diverse enzymes, mostly known to catalyse FA conjugation reactions [38]. The GSTs transport differ-ent molecules [38] imply that GSTCD might transport FA to the tissues and thus involved in the FA metabolism in sheep. This study found that genes playing roles in fatty acid-binding protein (FABPs) were deregulated in higher USFA samples. Fatty acid-binding proteins such as B-FABP or FABP7 are known to be involved in the intracellular transport of PUSFA [39]. FABPs are intracellular proteins involved in binding and intracellular trafficking of FA for metabolism and energy production [40].

## Biological function analysis for DEGs

Functional analysis showed that GO categories: biological processes, cellular components, and molecular functions were enriched in this study (Fig 3). The enriched biological processes identified were mainly related to cytokinesis, glycoprotein metabolic process, mitotic spindle,

protein N-linked glycosylation, acute inflammatory response, and regulation of developmental process. Mitotic spindle organization plays a role in FA metabolism and energy productionin mammalian cells [41]. Cellular components consisted of cell projection part, extracellular space, integral to plasma membrane, and proteinaceous extracellular matrix were significantly enriched by the DEGs. Among the cellular components, proteinaceous extracellular matrix plays a role in skeletal muscle development in wagyu cattle [42]. The molecular functions identified were mostly related to kinase inhibitor activity, growth factor binding, GTPase activity, carbohydrate binding. It has been reported that growth factor binding is associated with serum insulin-like growth factor binding, thus influence lipid composition [43]. Carbohydrate binding is an important factor that influences FA metabolism in rat [44].

A total of 11 significantly enriched KEGG pathways were identified for DEGs (Fig 4). Pathway analysis revealed that glycosaminoglycans biosynthesis- keratan sulphate (KS), adipokine signaling, galactose metabolism, endocrine and other factor-regulated calcium metabolism, mineral metabolism, and PPAR signaling pathways have important regulatory roles in FA metabolism in the liver tissues. Keratan sulphate plays a crucial role in cells growth, proliferation, and adhesion [45]. Adipokine signaling acts as a bridge between nutrition and obesity-related conditions [46]. Galactose metabolism is important for foetal and neonatal development as well as for adulthood [47]. Endocrine and other factor-regulated calcium metabolism, and mineral metabolism pathways are involved in intracellular mineral and calcium transportation, thus play roles in muscle muscle growth. Other important over-represented pathways in higher USFA group were phagosome and PPARs signaling pathway which were previously reported to be responsible for amino acid metabolism in cattle [16]. Several genes (APOA5, FABP7 and CPT1C) belonging to PPAR signaling pathway are identified in this study which could be involved in the FA metabolism in the seep. Berger and Moller [48] reported that PPARs are nuclear hormone receptors that are activated by FA and their derivatives, and regulate adipose tissue development and lipid metabolism in skeletal muscle. PPAR alpha is known to be involved in lipid metabolism in the liver and skeletal muscle, as well as in blood glucose uptake [49, 50]. The PPAR signaling pathway was identified as the most significantly over-represented pathway involved in FA composition in cattle using RNA-seq [16], suggesting that PPAR could have a key role in controlling FA metabolism in sheep.

## Regulatory hub genes of the hepatic transcriptome network

Regulatory hub genes of the hepatic transcriptome network identified several key genes including SOCS3, CBX6, MCM4, ITGB3, TGFBR2, GPRASP1, CELSR3, SDC3, SPOCK1, SEL1L and LEPR, which were upregulated in the liver tissues with higher USFA sheep (Fig 5A). The SOCS3 negatively regulates JAK2/STAT5a signaling, thus inhibits FA synthesis in cow [51]. ITGB3 gene affects marbling development by promoting lipid accumulation and facilitates hepatic insulin [52]. The potential downregulated Hub genes identified were ACTA2, GPRASP1, TPM2, TGM3, PTK6, and LTF (Fig 5B). ACTA gene controls muscle filaments and energy utilisation in muscle [53]. GPRASP1 is involved in Calcium (Ca2+) release by skeletal muscle [54]. We, therefore, speculated that the potential network hubs identified in this study might play important roles in the FA composition in sheep. The co-expression network illustrated that RACGAP1, MCM4, SDC3, CKAP2, RNASE6, PREX1, QSOX1, and FUT11 were the upregulated Hub genes (Fig 6A). RACGAP1 gene involved in oxidative functions in skeletal muscle cells [55]. QSOX1 gene is reported to be involved in meat quality, lipid metabolism, and cell apoptosis, and suggested to use as a biomarker for cattle breeding for superior meat quality [56]. The co-expression network illustrated that NRN1, TPM2, CDC42EP5, SSC5D, GPRASP1, and HRC were the downregulated Hub genes (Fig 6B). NRN1

gene was expressed in various mammalian tissues including lipid rafts of cell membrane [57]. TPM2 gene is reported to be involved in muscle marbling development and suggested to be a candidate gene for meat quality traits in cattle [58]. Although, most of the co-expression networks were individually involved in FA composition traits, however, they exert functions through participating in different directions which implies that the FA composition is influenced by gene expression changes, and it is a complex physiological process.

### Association between candidate markers and phenotypes

Selected polymorphisms within the APOA5, CFHR5, TFGBR2, and LEPR genes were found to be associated with the fatty acid composition phenotypes in this study (Table 6). The APOA5 is mapped on the ovine chromosome 15, which is an important factor for triglyceride rich lipoprotein (TLR) regulation [59]. A member of APO gene family, APOV1 also known as APOVL-DLII, is found to be down regulated in higher (UFA) sheep. This gene was previously reported to be associated with UFA in chicken [60]. Significant association between the variants in APOA5 gene and high triglyceride levels and FA composition have been previously documented in sheep [61, 62]. APOA5 is expressed in the liver, and controls VLDL binding (very low-density lipoprotein) to lipoprotein lipase (LPL) during FA synthesis in skeletal muscle and adipose tissue [63]. The CFHR5 is a 65 kDa plasma protein, binds with C3b, a C-reactive protein. Transforming growth factor beta receptor member familly 2 (TGBR2) is a member of the TGF-beta signaling pathway, which is involved in many cellular processes including cell growth, differentiation, and cellular homeostasis in animals [16]. The TGBPR2 gene is reported to be involved in myristoleic (C14: 1) FA metabolism [64]. Leptin receptor (LEPR) is an adipocytokine that regulates energy intake and uses in animals. Note, these polymorphisms are novel in sheep, and no association study with meat quality traits and FA compositions was conducted previously, so it is difficult to compare the results of this study with previous research. The LEPR was reported to be significantly associated with saturated FA, monounsaturated FA and polyunsaturated FA in pigs [1, 65]. The upregulation of LEPR in higher polyunsaturated FA group and significant association indicate that this gene and marker may control the FA metabolism in sheep. Therefore, it could be postulated that LEPR, as a putative candidate gene plays crucial role in regulating fatty acid composition and metabolism in sheep.

## Conclusion

The hepatic whole genome expression signature controlling unsaturated fatty acids (FA) levels in the sheep meat is, to our knowledge, deciphered for the first time. RNA-Seq provided a high-resolution map of transcriptional activities in the sheep liver tissue. The improvements in sheep genome annotations may lead to better coverage and detailed understanding of genomics controlling FA metabolism. This transcriptome analysis using RNA deep sequencing revealed potential candidate genes affecting FA composition and metabolism. This study suggested that candidate genes such as as APOA5, SLC25A30, GFPT1, LEPR, TGFBR2, FABP7, GSTCD, and CYP17A might be involved in the hepatic FA metabolism, thus control FA composition in muscle. Furthermore, number of SNPs were detected in the hepatic DEGs, and their associations with muscle FA compositions were validated. This transcriptome and polymorphism analyses using RNA Seq combined with association analysis showed potential candidate genes affecting FA composition and regulation in sheep. It is speculated that these polymorphisms could be used as markers for FA composition traits. However, further validation is required to confirm the effect of these genes and polymorphisms in other sheep populations.

## Materials and methods

### Animals and phenotypes

Tissue samples and phenotypes were collected from the Indonesian Javanese thin-tailed sheep. All sheep (n = 100) were slaughtered in PT Pramana Pangan Utama, IPB University, and used for phenotyping as well as for association analysis. Animal's breeding, rearing and management, growth performance, carcass and meat quality data were collected according to guidelines of the Indonesian performance test. Animals were slaughtered with an average age of 12 months, and 30 kg of liveweight in slaughterhouse, in accordance with the Indonesian Inspection Service procedures and was approved by the 'Institutional Animal Care and Use Committee (IACUC)" issued by IPB University (approval ID: 117–2018 IPB). Tissue samples from the longissimus muscle (at least 500g between the 12/13th ribs) of each animal (left half of the carcass) were removed for this study. Tissue samples from the longisimuss muscle and the liver were collected, frozen in liquid nitrogen immediately after slaughter and stored at -80˚C until used for RNA extraction. Similar tissue samples were collected and stored at -20˚C for FA analysis. Fatty acids (FA) compositions were determined for each sample using the extraction method regularly performed in our Lab following Folch et al. [66]. Briefly, muscle samples (~100 g) were grinded for FA composition. The lipids were extracted by homogenizing the samples with a chloroform and methanol (2:1) solution. NaCl at 1.5% was added so that the lipids were isolated. The isolated lipids were methylated, and the methyl esters were prepared from the extracted lipids with BF3-methanol (Sigma-Aldrich, St. Louis, MO, USA) and separated on a HP-6890N gas chromatograph (Hewlett-Packard, Palo Alto, CA, USA) as described previously [67]. Gas-chromatography/mass spectrometry (GC-MS) method was applied for the quantification of FA compositions [66, 67]. The average of USFA (MUSFA and PUSFA) and SFA value for these selected animals were 30.60 ± 10.12 and 39.73 ± 9.22 μg/g, respectively. Sheep having average USFA ≥45.59% μg/g and ≤25.84% μg/g were considered as higher-USFA (HUSFA) and lower-USFA (LUSFA) group, respectively (Table 1). In case of SFA, sheep having a SFA level ≤23.92% and ≥44.69% were considered as lower- and higher- SFA samples, respectively. However, for the transcriptome study, six sheep with divergently higher (n = 3) and lower (n = 3) USFA levels were selected from the total sheep (n = 100) population (Table 1). Total RNA was extracted from liver tissues using RNeasy Mini Kit according to the manufacturer's recommendations (Qiagen). Total RNA was treated using one-column RNase-Free DNase set (Promega), and quantified using a spectrophotometer (NanoDrop, ND8000, Thermo Scientific). RNA quality was assessed using an Agilent 2100 Bioanalyser and RNA Nano 6000 Labchip kit (Agilent Technologies).

### Library construction and sequencing

RNA integrity was verified by Agilent 2100 Bioanalyser® (Agilent, Santa Clara, CA, USA), where only samples with RIN > 7 were used for RNA deep sequencing. A total of 2 μg of RNA from each sample was used for library preparation according to the protocol described in Tru-Seq RNA Sample Preparation kit v2 guide (Illumina, San Diego, CA, USA). RNA deep sequencing technology was used to obtain the transcriptome expression. For this purpose, full-length cDNA library was constructed from 1 μg of RNA using the SMART cDNA Library Construction Kit (Clontech, USA), according to the manufacturer's instructions. Libraries of amplified RNA for each sample were prepared following the Illumina mRNA-Seq protocol. The prepared libraries were sequenced in an Illumina HiSeq 2500 as single-reads to 100 bp using 1 lane per sample on the same flow-cell (first sequencing run) at Macrogen Inc, South Korea. The sequencing data have been deposited in NCBI (Accession: PRJNA764003, ID: 764003). All sequences are analysed using the CASAVA v1.7 (Illumina, USA).

## Differential gene expression analysis

According to the FA concentration, animals were divided into two divergent phenotype value group (HUSFA and LUSFA) to identify differential expression genes (DEGs). The differential gene expression analysis was designed to contrast the differences in the expression of genes between two divergent sample group. The R package DESeq was used for the DEG analysis with raw count data [68]. The normalization procedure in DESeq handles the differences in the number of reads in each sample. For this purpose, DESeq first generates a fictitious reference sample with read counts defined as the geometric mean of all the samples. The read counts for each gene in each sample is divided by this geometric mean to obtain the normalized counts. To model the null distribution of computed data, DESeq follows an error model that uses a negative binomial distribution, with the variance and mean associated with regression. The method controls type-I error and provides good detection power [68]. After analysis using DESeq, DEGs were filtered based on $p$-adjusted value 0.05 and fold change $\geq$ 1.5 [69]. Additionally, the gene expression data was analysed using a Generalized Linear Model (GLM) function implemented in DESeq to calculate both within and between group deviances. As sanity checking and filtration step, we cross- matched the results from both analysis ($p$-adjusted $\leq$ 0.05 and fold change $\geq$ 1.5 criteria, and GLM analysis) and only those genes which appeared to be significant in both of the tests ($p$ value $\leq$ 0.05) were selected for further analysis.

## GO and pathways analysis

For biological interpretation of the DEGs, the GO and pathways enrichment analyses were performed using the NetworkAnlayst online tool [70]. For GO term enrichment, we used the GO database (http://geneontology.org/) and for pathways enrichment we used Kyoto Encyclopedia for Genes and Genomes (KEGG) database (https://www.genome.jp/kegg/pathway.html) incorporated in the NetworkAnlayst tool. The hypergeometric algorithm was applied for enrichment followed by Benjamini and Hochberg (H-B) [74] correction of multiple test.

## Network enrichment analyses

To identify the regulatory genes, the sub-network enrichment analysis was performed using the NetworkAnlayst online tool [70]. The tissue-specific protein-protein interactions (PPI) data from DifferetialNet Basha et al. [71] databases incorporated with NetworkAnalyst with medium percentile were used for the creation of liver specific PPI network. The orthologous human symbol of the DEGs were uploaded into the NetworkAnalyst to construct the liver tissue-specific PPI network. The default network created one bigger subnetwork "continent", and 14 smaller subnetwork "islands". All the islands contain only single seed gene; therefore, those were not considered further. For high performance visualization, the continent subnetwork was modified by using the minimize function of the tool. The network was depicted as nodes (circles representing genes) connected by edges (lines representing direct molecular interactions). Two topological measures such as degree (number of connections to other nodes) and betweenness (number of shortest paths going through the node) centrality were taken into account for detecting highly interconnected genes (hubs) of the network. Nodes having higher degree and betweenness were considered as potentially important network hubs in the cellular signal trafficking. In addition, liver specific genes co-expression networks were also constructed using the TCSBN database Lee et al. [72] incorporated into NetworkAnalyst tool.

## Quantitative Real Time PCR (qRT-PCR)

The cDNA was synthesised by reverse transcription PCR using 2 μg of total RNA, SuperScript II reverse transcriptase (Invitrogen) and oligo(dT)12 primer (Invitrogen). Gene specific primers for the qRT-PCR was designed by using the Primer3 software [73]. In each run, the 96-well microtiter plate was contained each cDNA sample, and no-template control. The qRT-PCR was conducted with the following program: 95˚C for 3 min, and 40 cycles: 95˚C for 15 s/60˚C for 45 s on the StepOne Plus qPCR system (Applied Biosystem). For each PCR reaction, 10 μl iTaqTM SYBR® Green Supermix with Rox PCR core reagents (Bio-Rad), 2 μl of cDNA (50 ng/μl) and an optimized amount of primers were mixed with ddH$_2$O to a final reaction volume of 20 μl per well. All samples were analysed twice (technical replication), and the geometric mean of the Ct values were further used for mRNA expression profiling. The housekeeping genes GAPDH and β-Actin were used for normalization of the target genes which were previously used for similar purpose in sheep tissues by our group [20]. The delta Ct (ΔCt) values was calculated as the difference between the target gene and geometric mean of the reference genes: (ΔCt = Ct$_{target}$−Ct$_{housekeeping}$ genes) as described in Silver *et al.* [74]. The final results were reported as the fold change calculated from delta Ct-values.

## Gene variation analysis

For gene variation analysis, SNP calls were performed on the mapping files generated by TopHat algorithm using 'samtools mpileup' command and associated algorithms [75]. Of the resulting variants, we selected the variants with a minimum Root Mean Square (RMS) mapping quality of 20 and a minimum read depth of 100 for further analyses. The selected variants were cross-checked against dbSNP database to identify mutations that had already studied. We also crosschecked and filtered the variants by the chromosomal positions of these variants against DEGs, and retained only those variants which mapped to DEG chromosome positions in order to find out the differentially expressed genes that also harboured sequence polymorphisms. By this way, we were able to isolate a handful of mutations that mapped to DEGs from many thousands of identified potential sequence polymorphisms. Furthermore, in order to understand whether these identified polymorphisms were segregated either in only one sample group (higher USFA and lower USFA) or in both groups (higher and lower USFA group), we calculated the read/coverage depth of these polymorphisms in all the samples [76]. The identified SNPs were classified as synonymous or non-synonymous using the GeneWise software (http://www.ebi.ac.uk/Tools/psa/genewise/ last accessed on 20.04.2021) by comparing between protein sequence and nucleotides incorporated SNP position [77].

## Validation of SNP and association study

For the validation of association study, a SNP in each of four highly polymorphic DEGs (APOA5, CFHR5, TGFBR2 and LEPR) as well as the genes to be played key role in the fatty acid metabolism were selected for association study (Table 6). A total 100 sheep were slaughtered, and the blood sample were taken for DNA extraction until we got a final concentration of 50 ng/ml DNA. The genotyping process were performed by PCR-RFLP (Polymerase Chain Reaction-Restriction Fragment Length Polymorphism) method. The PCR were performed in a 15 ml volume containing 1 ml of genomic DNA, 0.4 μl of primers, 6.1 μl of MyTaq HS Red Mix, 7.5 μl of nuclease water. The PCR product was checked on 1.5% agarose gel (Fischer Scientific Ltd) and digested by using the appropriate restriction enzyme. Digested PCR-RFLP products were resolved in 2% agarose gels. Effect of genotypes on fatty acid composition was performed with PROC GLM using SAS 9.2 (SAS Institute Inc, Cary, USA). Least square mean

values for the loci genotypes were compared by t-test, and p-values were adjusted by the Tukey-Kramer correction [78].

## Supporting information

**S1 Table. Differentially expressed genes with higher and lower fatty acid content in the liver of Javanese fat tailed sheep.**
(XLSX)

**S2 Table. List of genes involved in PPI network related to fatty acid metabolism in the liver of Javanese fat tailed sheep.**
(XLSX)

**S3 Table. List of genes involved in co-expression network related to fatty acid metabolism in the liver of Javanese fat tailed sheep.**
(XLSX)

**S4 Table. Total SNP detected by RNA-Seq in liver Javanese fat tailed sheep with higher and lower fatty acid composition.**
(XLSX)

**S5 Table. Genotype, allele frequencies and the chi-square test of selected SNPs validated using RFLP.**
(DOCX)

## Author Contributions

**Conceptualization:** Asep Gunawan, Muhammad Jasim Uddin.

**Data curation:** Asep Gunawan, Kasita Listyarini.

**Formal analysis:** Ratna Sholatia Harahap, Md. Aminul Islam.

**Funding acquisition:** Asep Gunawan.

**Investigation:** Jakaria, Katrin Roosita.

**Project administration:** Asep Gunawan, Kasita Listyarini.

**Resources:** Jakaria, Ismeth Inounu.

**Software:** Md. Aminul Islam.

**Supervision:** Asep Gunawan, Cece Sumantri, Muhammad Jasim Uddin.

**Validation:** Asep Gunawan, Katrin Roosita.

**Writing – original draft:** Asep Gunawan, Muhammad Jasim Uddin.

**Writing – review & editing:** Asep Gunawan, Cece Sumantri, Ismeth Inounu, Syeda Hasina Akter, Md. Aminul Islam, Muhammad Jasim Uddin.

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
