## [Decision Letter · Decision Letter 0]

3 Aug 2021

PONE-D-21-19113

Hepatic transcriptome analysis identifies genes, polymorphisms and pathways involved in the fatty acids metabolism in sheep

PLOS ONE

Dear Dr. Asep Gunawan,

Thank you for submitting your manuscript to PLOS ONE. After careful consideration, we feel that it has merit but does not fully meet PLOS ONE’s publication criteria as it currently stands. Therefore, we invite you to submit a revised version of the manuscript that addresses the points raised during the review process.

We look forward to receiving your revised manuscript.

Kind regards,

Islam Hamim, PhD

Academic Editor

PLOS ONE

Journal Requirements:

2. In your Methods section, please provide the name of the slaughterhouse where the animals were sacrificed

- https://www.sciencedirect.com/science/article/pii/S2452014419300123?via%3Dihub

In your revision ensure you cite all your sources (including your own works), and quote or rephrase any duplicated text outside the methods section. Further consideration is dependent on these concerns being addressed

"This work was supported by a project World Class Research (WCR) Number: 077/SP2H/LT/DRPM/2021 from the Ministry of Agriculture of the Republic of Indonesia."

"This work was supported by a project World Class Research (WCR) Number: 077/SP2H/LT/DRPM/2021 from the Ministry of Agriculture of the Republic of Indonesia."

Additional Editor Comments (if provided):

This is an interesting study, but the manuscript needs to be rewritten to reflect the reviewers' suggestions. I recommend that the authors make the major revisions that have been suggested by  reviewers.

Reviewers' comments:

Reviewer's Responses to Questions

**Comments to the Author**

1. Is the manuscript technically sound, and do the data support the conclusions?

Reviewer #1: Partly

Reviewer #2: Yes

Reviewer #3: Partly

2. Has the statistical analysis been performed appropriately and rigorously? 

Reviewer #1: N/A

Reviewer #2: Yes

Reviewer #3: No

3. Have the authors made all data underlying the findings in their manuscript fully available?

Reviewer #1: No

Reviewer #2: No

Reviewer #3: Yes

4. Is the manuscript presented in an intelligible fashion and written in standard English?

Reviewer #1: No

Reviewer #2: No

Reviewer #3: Yes

5. Review Comments to the Author

Reviewer #1: The study by Asep Gunawan et al. aims to elucidate genes and pathways involved in fatty acid metabolism using RNA sequencing technology to reveal differentially expressed genes in the liver tissues from sheep with high and low unsaturated fatty acid. In addition, the authors profile gene expression of putative new candidate genes for high and low unsaturated fatty acid and analyze its association with the phenotype under study. The paper has some merit in that it reports the association of DNA variants with high/low unsaturated fatty acid, following what is common practice in the study of candidate genes. Unfortunately, I have a number of concerns below detailed in no specific order.

1. The use of language makes the paper hard to understand. The manuscript needs thorough language editing; authors must provide certified proof of English language editing.

2. Abstract- #Line40, high USFA, and low SFA; and/or higher or lower USFA? in #LINE 43-44. please, choose the right word to describe the traits.

3. I am an animal geneticist and functional genomics analyst, so I think that the Introduction section does a poor job.

4. Table 1 is not clear….

#line920 (abMean value with different superscript letters in the same row differ significantly at P<0.05 ) ab-superscript is missing in Table 1, or is there no significant difference between the traits measured?

5. #Line 43-44, and #line 136-137---- These sentences are contracting and not clear. Authors need to be more specific in the choice of words used to describe high and low fatty acids.

6. #Table 3, I can’t see the list of up-and down-regulated genes as described by the authors. I would suggest an additional column representing up and down-regulated genes between sheep with high and low unsaturated fatty acids.

7. More than six pages of discussion seem too much, despite the authors didn’t discuss the causative mutations involved in regulating high and low unsaturated fatty acids in sheep.

8. The sequencing datasets from this study cannot be accessed. The authors should provide the accession number.

*******

Reviewer #2: I think this experiment is very meaningful. However, there are some major problems with the manuscript.

Figure 3, 4, 5, 6 not found in the manuscript, and Figures are not clear. Authors are encouraged to resubmit.

Reviewer #3: 1. Sheep having USFA >45.59 % μg/g and <25.84 % μg/g was considered as high-USFA (HUSFA) and low-USFA (LUSFA) group, respectively. But in the table 1, the mean USFA of HUSFA and LUSFA were 25.84% and 45.59%, respectively. So, the meaning of two sentences does not coincide.

2. A total 100 sheep were slaughtered and the blood samples were taken for DNA extraction for validation of SNP and association study. Weather the 100 sheep were the same as the 100 sheep for FA analysis? The authors did not make clear.

3. The average expression values of GAPDH and β-Actin was used to normalize the gene expression value. Why select these two genes? According to experiment test or other’s reports? The reason or criteria to select GAPDH and β-Actin should write clearly.

6. PLOS authors have the option to publish the peer review history of their article (what does this mean?). If published, this will include your full peer review and any attached files.

Reviewer #1: No

Reviewer #2: No

Reviewer #3: No

---

## [Author Response · Author response to Decision Letter 0]

18 Sep 2021

Response to Editor

Comment: PONE-D-21-19113

Hepatic transcriptome analysis identifies genes, polymorphisms and pathways involved in the fatty acids metabolism in sheep

PLOS ONE

Dear Dr. Asep Gunawan,

Thank you for submitting your manuscript to PLOS ONE. After careful consideration, we feel that it has merit but does not fully meet PLOS ONE’s publication criteria as it currently stands. Therefore, we invite you to submit a revised version of the manuscript that addresses the points raised during the review process.

• A rebuttal letter that responds to each point raised by the academic editor and reviewer(s). You should upload this letter as a separate file labelled 'Response to Reviewers'.

• A marked-up copy of your manuscript that highlights changes made to the original version. You should upload this as a separate file labelled 'Revised Manuscript with Track Changes'.

• An unmarked version of your revised paper without tracked changes. You should upload this as a separate file labelled 'Manuscript'.

We look forward to receiving your revised manuscript.

Kind regards,

Islam Hamim, PhD

Academic Editor

PLOS ONE

Response: We would like to thank the editor for marking our manuscript ‘interested’ and for detailed guidance to make the manuscript publishable. We would like to take this opportunity to thank the reviewers for their valuable time and comments. We do believe that these comments improve the manuscript significantly. We have responded all the comments from the reviewers point by point. 

Response to the journal office and /or Editor

Comment: Journal Requirements:

2. In your Methods section, please provide the name of the slaughterhouse where the animals were sacrificed

Response: Thank you for this point. We have added the name of slaughterhouse in the revised manuscript. Please see the Method ‘PT Pramana Pangan Utama’ which is located in IPB University (line no. 447-448). The file naming and necessary formatting have been done. Please see the revised manuscript with track changes.

Comment: 3. We noticed you have some minor occurrence of overlapping text with the following previous publication(s), which needs to be addressed:

- https://www.sciencedirect.com/science/article/pii/S2452014419300123?via%3Dihub

In your revision ensure you cite all your sources (including your own works), and quote or rephrase any duplicated text outside the methods section. Further consideration is dependent on these concerns being addressed.

Response: We appreciated the suggestions. The above-mentioned manuscript is published by our group (Gunawana A, Listyarinia K, Furqona A, Jakaria, Sumantri C, Akter SH and Uddin MJ. RNA deep sequencing reveals novel transcripts and pathways involved in the unsaturated fatty acid metabolism in chicken. Gene Reports, 15: 100370. 2019.) which was a similar study performed in chicken. In this manuscript we performed more vigorous analysis, as well as association studies for 4 SNPs in 4 different candidate genes and expression validation of 8 candidate genes using qRT-PCR. We ensure you that there is no overlapping. Note, the manuscript has been cited in the revised manuscript.

Comment: 4. We note that you have stated that you will provide repository information for your data at acceptance. Should your manuscript be accepted for publication, we will hold it until you provide the relevant accession numbers or DOIs necessary to access your data. If you wish to make changes to your Data Availability statement, please describe these changes in your cover letter and we will update your Data Availability statement to reflect the information you provide.

Response: Thanks. Please change the Data Availability’ statement. The data has been submitted to NCBI (Accession: PRJNA764003, ID: 764003). We apologies.

Comment: 5. Thank you for stating the following in the Acknowledgments Section of your manuscript: "This work was supported by a project World Class Research (WCR) Number: 077/SP2H/LT/DRPM/2021 from the Ministry of Agriculture of the Republic of Indonesia."

Please remove any funding-related text from the manuscript and let us know how you would like to update your Funding Statement. Currently, your Funding Statement reads as follows: "This work was supported by a project World Class Research (WCR) Number: 077/SP2H/LT/DRPM/2021 from the Ministry of Agriculture of the Republic of Indonesia."

Response: Please update the funding details in the ‘Funding Statement section of the online submission form’ on Author’s behalf. We appreciated your cooperation.

Comment: 6. We note that you have included the phrase “data not shown” in your manuscript. Unfortunately, this does not meet our data sharing requirements. PLOS does not permit references to inaccessible data. We require that authors provide all relevant data within the paper, Supporting Information files, or in an acceptable, public repository. Please add a citation to support this phrase or upload the data that corresponds with these findings to a stable repository (such as Figshare or Dryad) and provide and URLs, DOIs, or accession numbers that may be used to access these data. Or, if the data are not a core part of the research being presented in your study, we ask that you remove the phrase that refers to these data.

Response: The phrase ‘data not shown’ has been removed, please see the revised manuscript with track changes.

Comment: Additional Editor Comments (if provided):

This is an interesting study, but the manuscript needs to be rewritten to reflect the reviewers' suggestions. I recommend that the authors make the major revisions that have been suggested by reviewers.

Response: Thanks for the encouraging comment. We have edited the manuscript following reviewer’s comments. Please see the revised manuscript with track changes.

Response to reviewers’ comments

 Comments to the Author

1. Is the manuscript technically sound, and do the data support the conclusions?

Reviewer #1: Partly

Reviewer #2: Yes

Reviewer #3: Partly

2. Has the statistical analysis been performed appropriately and rigorously?

Reviewer #1: N/A

Reviewer #2: Yes

Reviewer #3: No

3. Have the authors made all data underlying the findings in their manuscript fully available?

Reviewer #1: No

Reviewer #2: No

Reviewer #3: Yes

4. Is the manuscript presented in an intelligible fashion and written in standard English?

Reviewer #1: No

Reviewer #2: No

Reviewer #3: Yes

5. Review Comments to the Author

Response: Thanks for the comments. We have edited the manuscript following reviewer’s comments. Please see the response below as well as the revised manuscript with track changes.

Response to reviewer #1

Comment: Reviewer #1: The study by Asep Gunawan et al. aims to elucidate genes and pathways involved in fatty acid metabolism using RNA sequencing technology to reveal differentially expressed genes in the liver tissues from sheep with high and low unsaturated fatty acid. In addition, the authors profile gene expression of putative new candidate genes for high and low unsaturated fatty acid and analyze its association with the phenotype under study. The paper has some merit in that it reports the association of DNA variants with high/low unsaturated fatty acid, following what is common practice in the study of candidate genes. Unfortunately, I have a number of concerns below detailed in no specific order.

Response: We would like to thank the reviewer #1 for his/her time and valuable comments. We have responded all the comments point by point. 

Comment: 1. The use of language makes the paper hard to understand. The manuscript needs thorough language editing; authors must provide certified proof of English language editing.

Response: We have revised the manuscript for language. An English native speaker (expert in Animal Science, Murdoch University, Australia) has volunteered in this regard.

Comment: 2. Abstract- #Line40, high USFA, and low SFA; and/or higher or lower USFA? in #LINE 43-44. please, choose the right word to describe the traits.

Response: Thanks for pointing this out. Since these are comparative traits so we have decided to use ‘higher’ and ‘lower’ rather than high and low USFA. Please see the revised manuscript with track changes.

Comment: 3. I am an animal geneticist and functional genomics analyst, so I think that the Introduction section does a poor job.

Response: Apologies for unwilling mistake. The introduction has been modified and improved. Please see the changes in the revised manuscript with track change. 

Comment: 4. Table 1 is not clear….#line920 (abMean value with different superscript letters in the same row differ significantly at P<0.05 ) ab-superscript is missing in Table 1, or is there no significant difference between the traits measured?

Response: Thank you for pointing this out. We apologize for the unwilling mistake. The superscript has been added in the revised Table 1.

Comment: 5. #Line 43-44, and #line 136-137-. These sentences are contracting and not clear. Authors need to be more specific in the choice of words used to describe high and low fatty acids.

Response: Apologies for creating confusion unwilling. The sentences have been revised in the revised manuscript (line no: 140-141).

Comment: 6. #Table 3, I can’t see the list of up-and down-regulated genes as described by the authors. I would suggest an additional column representing up and down-regulated genes between sheep with high and low unsaturated fatty acids.

Response: Thank you for this point. The table is getting large, so we have added a superscript (¥) and foot note in the table 3 to indicate up and down regulated gene. Please see the revised manuscript (at the end of Table 3). 

Comment: 7. More than six pages of discussion seem too much, despite the authors didn’t discuss the causative mutations involved in regulating high and low unsaturated fatty acids in sheep.

Response: The ‘Discussion’ has been modified and improved. Please see the changes in the revised manuscript with track change. Please note that this study includes vigorous sequence analysis, as well as association studies for 4 novel SNPs in 4 different candidate genes, and expression validation of 8 candidate genes using qRT-PCR. We have our best to explain and discussion the important findings. 

Comment: 8. The sequencing datasets from this study cannot be accessed. The authors should provide the accession number.

Response: The data has been submitted to NCBI (Accession: PRJNA764003, ID: 764003). We apologies.

Response to reviewer #2

Comment: Reviewer #2: I think this experiment is very meaningful. However, there are some major problems with the manuscript.

Figure 3, 4, 5, 6 not found in the manuscript, and Figures are not clear. Authors are encouraged to resubmit.

Response: Thank you so much for your time and encouraging comment. We have submitted all the figures separately along with manuscript according to the Plos One’s Author instructions. We have double checked the figures quality. We apologies for the issue.

Response to reviewer #3

Comment: Reviewer #3: 1. Sheep having USFA >45.59 % μg/g and <25.84 % μg/g was considered as high-USFA (HUSFA) and low-USFA (LUSFA) group, respectively. But in the table 1, the mean USFA of HUSFA and LUSFA were 25.84% and 45.59%, respectively. So, the meaning of two sentences does not coincide.

Response: Thank you so much for your time and encouraging comment. We apologies for the unwilling mistake. The statement has been revised and corrected. Please see the revised manuscript (line no: 468-470): ‘Sheep having USFA ≥45.59 % μg/g and ≤25.84 % μg/g was considered as higher-USFA (HUSFA) and lower-USFA (LUSFA) group, respectively’. 

Comment: 2. A total 100 sheep were slaughtered, and the blood samples were taken for DNA extraction for validation of SNP and association study. Weather the 100 sheep were the same as the 100 sheep for FA analysis? The authors did not make clear.

Response: We appreciate for the comments. The same 100 sheep were used for both association study and fatty acid analysis. The statement has been made clear in the revised manuscript with track change (please see line no. 446-447). 

Comment: 3. The average expression values of GAPDH and β-Actin was used to normalize the gene expression value. Why select these two genes? According to experiment test or other’s reports? The reason or criteria to select GAPDH and β-Actin should write clearly.

Response: We appreciated for this point. We have selected these two housekeeping genes based on our previous report conducted with muscle tissue in sheep (Gunawan et al, 2018) published in Gene, 676: 86-94. 

Reference

Gunawan, A., Jakaria, K Listyarini, A Furqon, C Sumantri, SH Akter, MJ Uddin. 2018. Transcriptome signature of liver tissue with divergent mutton odor and flavour using RNA deep sequencing. Gene. 676: 86-94.

---

## [Decision Letter · Decision Letter 1]

8 Oct 2021

PONE-D-21-19113R1Hepatic transcriptome analysis identifies genes, polymorphisms and pathways involved in the fatty acids metabolism in sheepPLOS ONE

Dear Dr. Gunawan,

Thank you for submitting your manuscript to PLOS ONE. After careful consideration, we feel that it has merit but does not fully meet PLOS ONE’s publication criteria as it currently stands. Therefore, we invite you to submit a revised version of the manuscript that addresses the points raised during the review process.

The Reviewers have 

We look forward to receiving your revised manuscript.

Kind regards,

Martina Zappaterra

Academic Editor

PLOS ONE

Journal Requirements:

Additional Editor Comments (if provided):

Reviewers have evaluated positively the manuscript. However, in order to avoid repetition and grammatical errors, authors are encouraged to have their manuscript professionally edited in English.

Reviewers' comments:

Reviewer's Responses to Questions

**Comments to the Author**

1. If the authors have adequately addressed your comments raised in a previous round of review and you feel that this manuscript is now acceptable for publication, you may indicate that here to bypass the “Comments to the Author” section, enter your conflict of interest statement in the “Confidential to Editor” section, and submit your "Accept" recommendation.

Reviewer #1: (No Response)

Reviewer #3: All comments have been addressed

2. Is the manuscript technically sound, and do the data support the conclusions?

Reviewer #1: Yes

Reviewer #3: Yes

3. Has the statistical analysis been performed appropriately and rigorously? 

Reviewer #1: N/A

Reviewer #3: Yes

4. Have the authors made all data underlying the findings in their manuscript fully available?

Reviewer #1: Yes

Reviewer #3: Yes

5. Is the manuscript presented in an intelligible fashion and written in standard English?

Reviewer #1: (No Response)

Reviewer #3: Yes

6. Review Comments to the Author

Reviewer #1: Dear Authors, I appreciate your efforts to improve your manuscript and your decision to publish it in PLoS One. However, in order to avoid repetition and grammatical errors, authors must have their manuscript professionally edited in English.

These are just a few of the grammatical errors discovered....:

In L43 delete “a” . From sheep (n=100)....(Plural form)

L109 "analysis" In addition, gene polymorphism and association analyses......(Plural form)....

Reviewer #3: (No Response)

7. PLOS authors have the option to publish the peer review history of their article (what does this mean?). If published, this will include your full peer review and any attached files.

Reviewer #1: No

Reviewer #3: No

---

## [Author Response · Author response to Decision Letter 1]

17 Oct 2021

Response to academic editor

Comment: PONE-D-21-19113R1

Hepatic transcriptome analysis identifies genes, polymorphisms and pathways involved in the fatty acids metabolism in sheep

PLOS ONE

Dear Dr. Gunawan,

Thank you for submitting your manuscript to PLOS ONE. After careful consideration, we feel that it has merit but does not fully meet PLOS ONE’s publication criteria as it currently stands. Therefore, we invite you to submit a revised version of the manuscript that addresses the points raised during the review process.

Please submit your revised manuscript by Nov 22 2021 11:59PM

We look forward to receiving your revised manuscript.

Kind regards,

Martina Zappaterra

Academic Editor

PLOS ONE

Response: We would like to thanks the editor and reviewers for marking the manuscript as ‘minor-revision’ and for their valuable time and comments. We do believe that these comments have improved the quality of the manuscript. English has been double checked with native English reader expert in animal science. All changes and corrections could be found in the revised manuscript with track change. 

Comment: Please review your reference list to ensure that it is complete and correct. If you have cited papers that have been retracted, please include the rationale for doing so in the manuscript text, or remove these references and replace them with relevant current references. Any changes to the reference list should be mentioned in the rebuttal letter that accompanies your revised manuscript. If you need to cite a retracted article, indicate the article’s retracted status in the References list and also include a citation and full reference for the retraction notice.

Response: Recent and appropriate references have been cited in this manuscript. We were careful to give credit to the original articles. All the references have been double checked both in the text and in the reference section. The references style has been corrected following journal style and according to author instructions. All changes and corrections could be found in the revised manuscript with track change. 

Comment: Additional Editor Comments (if provided):

Reviewers have evaluated positively the manuscript. However, in order to avoid repetition and grammatical errors, authors are encouraged to have their manuscript professionally edited in English.

Response: English has been double checked with native English speaker expert in animal science. All changes and corrections could be found in the revised manuscript with track change. 

Response to Reviewers' comments

Reviewer's Responses to Questions

Comment: Comments to the Author

1. If the authors have adequately addressed your comments raised in a previous round of review and you feel that this manuscript is now acceptable for publication, you may indicate that here to bypass the “Comments to the Author” section, enter your conflict of interest statement in the “Confidential to Editor” section, and submit your "Accept" recommendation.

Reviewer #1: (No Response)

Reviewer #3: All comments have been addressed

Comment: 2. Is the manuscript technically sound, and do the data support the conclusions?

Reviewer #1: Yes

Reviewer #3: Yes

Comment: 3. Has the statistical analysis been performed appropriately and rigorously?

Reviewer #1: N/A

Reviewer #3: Yes

Comment: 4. Have the authors made all data underlying the findings in their manuscript fully available?

Reviewer #1: Yes

Reviewer #3: Yes

Comment: 5. Is the manuscript presented in an intelligible fashion and written in standard English?

Reviewer #1: (No Response)

Reviewer #3: Yes

Comment: 6. Review Comments to the Author

Reviewer #1: Dear Authors, I appreciate your efforts to improve your manuscript and your decision to publish it in PLoS One. However, in order to avoid repetition and grammatical errors, authors must have their manuscript professionally edited in English.

These are just a few of the grammatical errors discovered....:

In L43 delete “a” . From sheep (n=100)....(Plural form)

L109 "analysis" In addition, gene polymorphism and association analyses......(Plural form)....

Reviewer #3: (No Response)

Response: We thanks the reviewer for pointing this out. We apologies for unwilling grammatical errors. English has been double checked with native English speaker expert in animal science. All changes and corrections could be found in the revised manuscript with track change. 

Comment: 7. PLOS authors have the option to publish the peer review history of their article (what does this mean?). If published, this will include your full peer review and any attached files.

Do you want your identity to be public for this peer review? For information about this choice, including consent withdrawal, please see our Privacy Policy.

Reviewer #1: No

Reviewer #3: No

Response: We would like to thank the editor and reviewers for their valuable time and comments. We do believe that these comments have improved the quality of the manuscript. The revision has been submitted according to the journal instructions.

---

## [Editor Report · Decision Letter 2]

12 Nov 2021

Hepatic transcriptome analysis identifies genes, polymorphisms and pathways involved in the fatty acids metabolism in sheep

PONE-D-21-19113R2

Dear Dr. Gunawan,

We’re pleased to inform you that your manuscript has been judged scientifically suitable for publication and will be formally accepted for publication once it meets all outstanding technical requirements.

Kind regards,

Martina Zappaterra

Academic Editor

PLOS ONE
---

## [Editor Report · Acceptance letter]

15 Dec 2021

PONE-D-21-19113R2 

Hepatic transcriptome analysis identifies genes, polymorphisms and pathways involved in the fatty acids metabolism in sheep 

Dear Dr. Gunawan:

I'm pleased to inform you that your manuscript has been deemed suitable for publication in PLOS ONE. Congratulations! Your manuscript is now with our production department. 

Kind regards, 

on behalf of

Dr. Martina Zappaterra 

Academic Editor

PLOS ONE